# Estimating and Evaluating Regression Predictive Uncertainty in Deep Object Detectors

**Ali Harakeh**
Institute for Aerospace Studies
University of Toronto
`ali.harakeh@utoronto.ca`

**Steven L. Waslander**
Institute for Aerospace Studies
University of Toronto
`stevenw@utias.utoronto.ca`

## Abstract

Predictive uncertainty estimation is an essential next step for the reliable deployment of deep object detectors in safety-critical tasks. In this work, we focus on estimating predictive distributions for bounding box regression output with variance networks. We show that in the context of object detection, training variance networks with negative log likelihood (NLL) can lead to high entropy predictive distributions regardless of the correctness of the output mean. We propose to use the energy score as a non-local proper scoring rule and find that when used for training, the energy score leads to better calibrated and lower entropy predictive distributions than NLL. We also address the widespread use of non-proper scoring metrics for evaluating predictive distributions from deep object detectors by proposing an alternate evaluation approach founded on proper scoring rules. Using the proposed evaluation tools, we show that although variance networks can be used to produce high quality predictive distributions, adhoc approaches used by seminal object detectors for choosing regression targets during training do not provide wide enough data support for reliable variance learning. We hope that our work helps shift evaluation in probabilistic object detection to better align with predictive uncertainty evaluation in other machine learning domains. Code for all models, evaluation, and datasets is available at: https://github.com/asharakeh/probdet.git.

## 1 Introduction

Deep object detectors are being increasingly deployed as perception components in safety critical robotics and automation applications. For reliable and safe operation, subsequent tasks using detectors as sensors require meaningful predictive uncertainty estimates correlated with their outputs. As an example, overconfident incorrect predictions can lead to non-optimal decision making in planning tasks, while underconfident correct predictions can lead to under-utilizing information in sensor fusion. This paper investigates probabilistic object detectors, extensions of standard object detectors that estimate predictive distributions for output categories and bounding boxes simultaneously.

This paper aims to identify the shortcomings of recent trends followed by state-of-the-art probabilistic object detectors, and provides theoretically founded solutions for identified issues. Specifically, we observe that the majority of state-of-the-art probabilistic object detectors methods (Feng et al., 2018a; Le et al., 2018; Feng et al., 2018b; He et al., 2019; Kraus & Dietmayer, 2019; Meyer et al., 2019; Choi et al., 2019; Feng et al., 2020; He & Wang, 2020; Harakeh et al., 2020; Lee et al., 2020) build on deterministic object detection backends to estimate bounding box predictive distributions by modifying such backends with variance networks (Detlefsen et al., 2019). The mean and variance of bounding box predictive distributions estimated using variance networks are then learnt using negative log likelihood (NLL). It is also common for these methods to use non-proper scoring rules such as the mean Average Precision (mAP) when evaluating the quality of their output predictive distributions.

**Pitfalls of NLL** We show that under standard training procedures used by common object detectors, using NLL as a minimization objective results in variance networks that output high entropy distributions regardless of the correctness of an output bounding box. We address this issue by using

the Energy Score (Gneiting & Raftery, 2007), a distance-sensitive proper scoring rule based on energy statistics (Székely & Rizzo, 2013), as an alternative for training variance networks. We show that predictive distributions learnt with the energy score are lower entropy, better calibrated, and of higher quality when evaluated using proper scoring rules.

**Pitfalls of Evaluation** We address the widespread use of non-proper scoring rules for evaluating probabilistic object detectors by providing evaluation tools based on well established proper scoring rules (Gneiting & Raftery, 2007) that are only minimized if the estimated predictive distribution is equal to the true target distribution, for both classification and regression. Using the proposed tools, we benchmark probabilistic extensions of three common object detection architectures on in-distribution, shifted, and out-of-distribution data. Our results show that variance networks can differentiate between in-distribution, shifted, and out-of-distribution data using their predictive entropy. We find that ad-hoc approaches used by seminal object detectors for choosing their regression targets during training do not provide a wide enough data support for reliable learning in variance networks. Finally, we provide clear recommendations in Sec. 5 to avoid the pitfalls described above.

## 2 RELATED WORK

**Estimating predictive distributions** with deep neural networks has long been a topic of interest for the research community. Bayesian Neural Networks (BNNs) (MacKay, 1992) quantify predictive uncertainty by approximating a posterior distribution over a set of network parameters given a predefined prior distribution. Variance networks (Nix & Weigend, 1994) capture predictive uncertainty (Kendall & Gal, 2017) by estimating the mean and variance of every output through separate neural network branches, and are usually trained using maximum likelihood estimation (Detlefsen et al., 2019). Deep ensembles (Lakshminarayanan et al., 2017) train multiple copies of the variance networks from different network initializations to estimate predictive distributions from output sample sets. Monte Carlo (MC) Dropout (Gal & Ghahramani, 2016) provides predictive uncertainty estimates based on output samples generated by activating dropout layers at test time. We refer the reader to the work of Detlefsen et al. (2019) for an in depth comparison of the performance of variance networks, BNNs, Ensembles, and MC dropout on regression tasks. We find variance networks to be the most scalable of these methods for the object detection task. Finally, we do not distinguish between *aleatoric* and *epistemic* uncertainty as is done in Kendall & Gal (2017), instead focusing on *predictive uncertainty* (Detlefsen et al., 2019) which reflects both types.

**State-of-the-art probabilistic object detectors** model predictive uncertainty by adapting the work of Kendall & Gal (2017) to state-of-the-art object detectors. Standard detectors are extended with a variance network, usually referred to as the variance regression head, alongside the mean bounding box regression head and the resulting network is trained using NLL (Feng et al., 2018a; Le et al., 2018; He et al., 2019; Lee et al., 2020; Feng et al., 2020; He & Wang, 2020). Some approaches combine the variance networks with dropout (Feng et al., 2018b; Kraus & Dietmayer, 2019) and use Monte Carlo sampling at test time. Others (Meyer et al., 2019; Choi et al., 2019; Harakeh et al., 2020) make use of the output predicted variance by modifying the non-maximum suppression post-processing stage. Such modifications are orthogonal to the scope of the paper. It is important to note that a substantial portion of existing probabilistic object detectors focus on non-proper scoring metrics such as the mAP and calibration errors to evaluate the quality of their predictive distributions. More recent methods (Harakeh et al., 2020; He & Wang, 2020) use the probability-based detection quality (PDQ) proposed by Hall et al. (2020) for evaluating probabilistic object detectors, which can also be shown to be non-proper (See appendix D.3). Instead, we combine the error decomposition proposed by Hoiem et al. (2012) with well established proper scoring rules (Gneiting & Raftery, 2007) to evaluate probabilistic object detectors.

## 3 LEARNING BOUNDING BOX DISTRIBUTIONS WITH PROPER SCORING RULES

### 3.1 NOTATION AND PROBLEM FORMULATION

Let $\mathbf{x} \in \mathbb{R}^m$ be a set of $m$-dimensional features, $\mathbf{y} \in \{1, \dots, K\}$ be classification labels for $K$-way classification, and $\mathbf{z} \in \mathbb{R}^d$ be bounding box regression targets associated with object instances in the

scene. Given a training dataset $\mathcal{D} = \{(\boldsymbol{x}_n, y_n, \boldsymbol{z}_n)\}_{n=1}^N$ of $N$ i.i.d samples from a true joint conditional probability distribution $p^*(\mathbf{y}, \mathbf{z}|\mathbf{x}) = p^*(\mathbf{z}|\mathbf{y}, \mathbf{x})p^*(\mathbf{y}|\mathbf{x})$, we use neural networks with parameter vector $\boldsymbol{\theta}$ to model $p_{\boldsymbol{\theta}}(\mathbf{z}|\mathbf{y}, \mathbf{x})$ and $p_{\boldsymbol{\theta}}(\mathbf{y}|\mathbf{x})$. $p_{\boldsymbol{\theta}}(\mathbf{z}|\mathbf{y}, \mathbf{x})$ is fixed to be a multivariate Gaussian distribution $\mathcal{N}(\boldsymbol{\mu}(\mathbf{x}, \boldsymbol{\theta}), \boldsymbol{\Sigma}(\mathbf{x}, \boldsymbol{\theta}))$, and $p_{\boldsymbol{\theta}}(\mathbf{y}|\mathbf{x})$ as a categorical distribution $\text{Cat}(p_1(\mathbf{x}, \boldsymbol{\theta}), \ldots, p_K(\mathbf{x}, \boldsymbol{\theta}))$. Unless mentioned otherwise, $\mathbf{z} \in \mathbb{R}^4$ and is represented as $(u_{\min}, v_{\min}, u_{\max}, v_{\max})$ where the $(u_{\min}, v_{\min})$, $(u_{\max}, v_{\max})$ are the pixel coordinates of the top-left and bottom-right bounding box corners respectively. Throughout this work, we denote random variables with bold characters, and the associated ground truth instances of random variables that are realized in the dataset $\mathcal{D}$ are italicized.

## 3.2 PROPER SCORING RULES

Let $\mathbf{a}$ be either $\mathbf{y}$ or $\mathbf{z}$. A scoring rule is a function $S(p_{\boldsymbol{\theta}}, (\boldsymbol{a}, \boldsymbol{x}))$ that assigns a numerical value to the quality of the predictive distribution $p_{\boldsymbol{\theta}}(\mathbf{a}|\mathbf{x})$ given the actual event that materialized $\boldsymbol{a}|\boldsymbol{x} \sim p^*(\mathbf{a}|\boldsymbol{x})$, *where a lower value indicates better quality*. With slight abuse of notation, let $S(p_{\boldsymbol{\theta}}, p^*)$ also refer to the expected value of $S(p_{\boldsymbol{\theta}}, .)$, then a scoring rule is said to be *proper* if $S(p^*, p^*) \leq S(p_{\boldsymbol{\theta}}, p^*)$, with equality if and only if $p_{\boldsymbol{\theta}} = p^*$, meaning that the actual data generating distribution is assigned the lowest possible score value (Gneiting & Raftery, 2007). We provide a more formal definition of proper scoring rules in Appendix D.2. Beyond the notion of proper, scoring rules can be further divided into *local* and *non-local* rules. Local scoring rules evaluate a predictive distribution based on its value only at the true target, whereas non-local rules take into account other characteristics of the predictive distribution. As an example, distance-sensitive non-local proper scoring rules reward predictive distributions that assign probability mass to the vicinity of the true target, even if not exactly at that target. Lakshminarayanan et al. (2017) noted the utility of using proper scoring rules as neural network loss functions for learning predictive distributions.

Predictive uncertainty for classification tasks has been extensively studied in recent literature (Ovadia et al., 2019; Ashukha et al., 2020). We find commonly used proper scoring rules such as NLL and the Brier score (Brier, 1950) to be satisfactory to learn and evaluate categorical predictive distributions for probabilistic object detectors. On the other hand, regression tasks have overwhelmingly relied on a single proper scoring rule, the negative log likelihood (Kendall & Gal, 2017; Lakshminarayanan et al., 2017; Detlefsen et al., 2019). NLL is a local scoring rule and should be satisfactory if used to evaluate pure inference problems (Bernardo, 1979). In addition, the choice of a proper scoring rule should not matter as asymptotically, the true parameters of $p_{\boldsymbol{\theta}}$ should be recovered by minimizing any proper scoring rule (Gneiting & Raftery, 2007) during training. Unfortunately, object detection does not conform to these idealized assumptions, we show in the next section the pitfalls of using NLL for learning and evaluating bounding box predictive distributions. We also explore the Energy Score (Gneiting & Raftery, 2007), a proper and non-local scoring rule as an alternative for learning and evaluating multivariate Gaussian predictive distributions.

## 3.3 NEGATIVE LOG LIKELIHOOD AS A SCORING RULE

For a multivariate Gaussian, the NLL can be written as:

$$\text{NLL} = \frac{1}{2N} \sum_{n=1}^N (\boldsymbol{z}_n - \boldsymbol{\mu}(\boldsymbol{x}_n, \boldsymbol{\theta}))^\mathsf{T} \boldsymbol{\Sigma}(\boldsymbol{x}_n, \boldsymbol{\theta})^{-1} (\boldsymbol{z}_n - \boldsymbol{\mu}(\boldsymbol{x}_n, \boldsymbol{\theta})) + \log \det \boldsymbol{\Sigma}(\boldsymbol{x}_n, \boldsymbol{\theta}), \quad (1)$$

where $N$ is the size of the dataset $\mathcal{D}$. NLL is the only proper scoring rule that is also local (Bernardo, 1979), with higher values implying a worse predictive density quality *at the true target value*. When used as a loss to minimize, the first term of NLL encourages increasing the entropy of the predictive distribution as the mean estimate diverges from the true target value. The log determinant regularization term has a contrasting effect, penalizing high entropy distributions and preventing a zero loss from infinitely high uncertainty predictions at all data points (Kendall & Gal, 2017).

It has been shown by Machete (2013) that NLL prefers predictive densities that are less informative, penalizing overconfidence even when the probability mass is concentrated on a likely outcome. We demonstrate the relevance of this property to object detection with a simple toy example. Bounding box results from the state-of-the-art object detector DETR (Carion et al., 2020), trained to achieve a competitive mAP of $42\%$ on the COCO validation split, are assigned a single mock multivariate Gaussian probability distribution $\mathcal{N}(\boldsymbol{\mu}(\boldsymbol{x}_n, \boldsymbol{\theta}), \sigma\boldsymbol{I})$, where $\sigma\boldsymbol{I}$ is a $4 \times 4$ isotropic covariance matrix

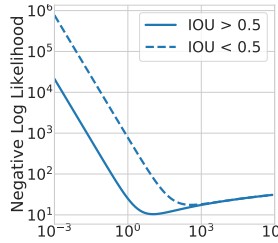 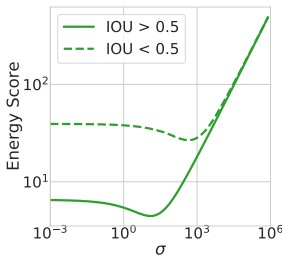 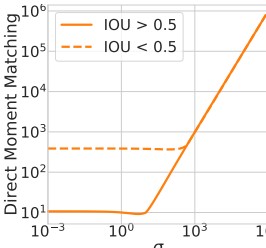

Figure 1: A toy example showing values of NLL (Blue), ES (Green) and DMM (Orange) plotted against the parameter $\sigma$ of an isotropic covariance matrix $\sigma \boldsymbol{I}$, when assigned to low error (Solid Lines) and high error (Dashed Lines) detection outputs from DETR.

with $\sigma$ as a variable parameter. We split the detection results into a high error set with an IOU $<=$ 0.5, and a low error set with an IOU $> 0.5$, where the IOU is determined as the maximum IOU with any ground truth bounding box in the scene. We plot the value of the NLL loss of both high error and low error detection instances in Figure 1, where NLL is estimated at values of $\sigma$ between $[10^{-2}, 10^5]$. As expected, the variance value that minimizes NLL for low error detections is two orders of magnitude lower than for high error detections. What is more interesting is the behavior of NLL away from its minimum for both low error and high error detections. NLL is seen to penalize lower entropy distributions (smaller values of $\sigma$) more severely than higher entropy distributions, a property that is shown to be detrimental for training variance networks in Section 4.

## 3.4 THE ENERGY SCORE (ES)

The energy score is a *strictly proper and non-local* (Gneiting et al., 2008) scoring rule used to assess probabilistic forecasts of multivariate quantities. The energy score and can be written as:

$$\text{ES} = \frac{1}{N} \sum_{n=1}^{N} \left( \frac{1}{M} \sum_{i=1}^{M} ||\mathbf{z}_{n,i} - \boldsymbol{z}_n|| - \frac{1}{2M^2} \sum_{i=1}^{M} \sum_{j=1}^{M} ||\mathbf{z}_{n,i} - \mathbf{z}_{n,j}|| \right), \quad (2)$$

where $\boldsymbol{z}_n$ is the ground truth bounding box, and $\mathbf{z}_{n,i}$ is the i[th] i.i.d sample from $\mathcal{N}(\boldsymbol{\mu}(\boldsymbol{x}_n, \boldsymbol{\theta}), \boldsymbol{\Sigma}(\boldsymbol{x}_n, \boldsymbol{\theta}))$.

The energy score is derived from the energy distance (Rizzo & Székely, 2016)[1], a maximum mean discrepancy metric (Sejdinovic et al., 2013) that measures distance between distributions of random vectors. Beyond its theoretical appeal, the energy score has an efficient Monte-Carlo approximation (Gneiting et al., 2008) for multivariate Gaussian distributions, written as:

$$\text{ES} = \frac{1}{N} \sum_{n=1}^{N} \left( \frac{1}{M} \sum_{i=1}^{M} ||\mathbf{z}_{n,i} - \boldsymbol{z}_n|| - \frac{1}{2(M-1)} \sum_{i=1}^{M-1} ||\mathbf{z}_{n,i} - \mathbf{z}_{n,i+1}|| \right), \quad (3)$$

which requires only a single set of $M$ samples to be drawn from $\mathcal{N}(\boldsymbol{\mu}(\boldsymbol{x}_n, \boldsymbol{\theta}), \boldsymbol{\Sigma}(\boldsymbol{x}_n, \boldsymbol{\theta}))$ for every object instance in the minibatch. For training our object detectors, we find that setting $M$ to be equal to 1000 allows us to compute the approximation in equation 3 with very little computational overhead. We also use the value of 1000 for $M$ when using the energy score as an evaluation metric.

The energy score has been previously used successfully as an optimization objective in DISCO Nets (Bouchacourt et al., 2016) to train neural networks that output samples from a posterior probability distribution, and to train generative adversarial networks (Bellemare et al., 2017). Unlike DISCO nets, we use the energy score to learn parametric distributions with differentiable sampling.

Since the energy distance is non-local, it favors distributions that place probability mass near the ground truth target value, even if not exactly at that target. Going back to the toy example in Figure 1, ES is shown to be minimized at similar values to the NLL, not a surprising observation given that both are proper scoring rules. Unlike NLL, ES penalizes high entropy distributions more severely than low entropy ones. In the next section, we observe the effects of this property when using

---

[1]See Appendix E.

the energy score as a loss, leading to better calibrated, lower entropy predictive distributions when compared to NLL.

## 3.5 Direct Moment Matching (DMM)

A final scoring rule we consider in our experiments for learning predictive distributions is the Direct Moment Matching (DMM) (Feng et al., 2020):

$$\text{DMM} = \frac{1}{N}\sum_{n=1}^{N}||\mathbf{z}_n - \boldsymbol{\mu}(\boldsymbol{x}_n, \boldsymbol{\theta})||_p + ||\boldsymbol{\Sigma}(\boldsymbol{x}_n, \boldsymbol{\theta}) - (\mathbf{z}_n - \boldsymbol{\mu}(\boldsymbol{x}_n, \boldsymbol{\theta}))(\mathbf{z}_n - \boldsymbol{\mu}(\boldsymbol{x}_n, \boldsymbol{\theta}))^\mathsf{T}||_p, \quad (4)$$

where $||.||_p$ is a $p$ norm. DMM was previously proposed by Feng et al. (2020) as an auxiliary loss to calibrate 3D object detectors using a multi-stage training procedure. DMM matches the mean and covariance matrix of the predictive distribution to sample statistics obtained using the predicted and true target values. DMM is not a proper scoring rule, we show this property with an example. If $(\mathbf{z}_n - \boldsymbol{\mu}(\boldsymbol{x}_n, \boldsymbol{\theta}))$ is a vector of zeros, DMM is minimized only if all entries of $\boldsymbol{\Sigma}(\boldsymbol{x}_n, \boldsymbol{\theta})$ are $0$ regardless of the actual covariance of the data generating distribution. We use results from variance networks trained with DMM to discuss the pitfalls of only relying on distance-sensitive proper scoring rules for evaluation.

## 4 Experiments And Results

For our experiments, we extend three common object detection methods, DETR (Carion et al., 2020), RetinaNet (Lin et al., 2017) and FasterRCNN (Ren et al., 2015) with variance networks to estimate the parameters of a multivariate Gaussian bounding box predictive distribution. We use cross-entropy to learn categorical predictive distributions and NLL, DMM, or ES to learn bounding box predictive distributions, resulting in 9 architecture/regression loss combinations. All networks have open source code and well established hyperparameters, which are *used as is* whenever possible. Our probabilistic extensions closely match the mAP reported in the original implementation of their deterministic counterparts (See Fig. F.5). The value of $p$ used for computing DMM in equation equation 4 is determined according to original norm loss used for bounding box regression in each deterministic backend. For FasterRCNN and RetinaNet, we use the smooth L1 loss, while for DETR, we use the L1 norm. Additional details on model training and inference implementations can be found in Appendix A.

All probabilistic object detectors are trained on the COCO (Lin et al., 2014) training data split. For testing, the COCO validation dataset is used as in-distribution data. Following recent recommendations for evaluating the quality of predictive uncertainty estimates (Ovadia et al., 2019), we also test our probabilistic object detectors on shifted data distributions. We construct 3 distorted versions of the COCO validation dataset (C1, C3, and C5) by applying 18 different image corruptions introduced by Hendrycks & Dietterich (2019) at increasing intensity levels $[1, 3, 5]$. To test on natural dataset shift, we use OpenImages data (Kuznetsova et al., 2020) to create a shifted dataset with the same categories as COCO and an out-of-distribution dataset that contain none of the $80$ categories found in COCO. More details on these datasets and their construction can be found in Appendix B.

The three deterministic backends are chosen to represent one-stage (RetinaNet), two-stage (Faster-RCNN), and the recently proposed set-based (DETR) object detectors. In addition, the implementation of DETR [2], RetinaNet, and FasterRCNN models is publicly available under the *Detectron2* (Wu et al., 2019) object detection framework, with hyperparameters optimized to produce the best detection results for the COCO dataset.

### 4.1 Evaluating Predictive Distributions With Proper Scoring Rules

Following the work of Hoiem et al. (2012), we partition output object instances into four categories: true positives, duplicates, localization errors, and false positives based on their IOU with ground truth instances in a scene. False positives are defined as detection instances with an IOU $\leq 0.1$ with any ground truth object instance in the image frame, whereas localization errors are detection

---

[2]https://github.com/facebookresearch/detr/tree/master/d2

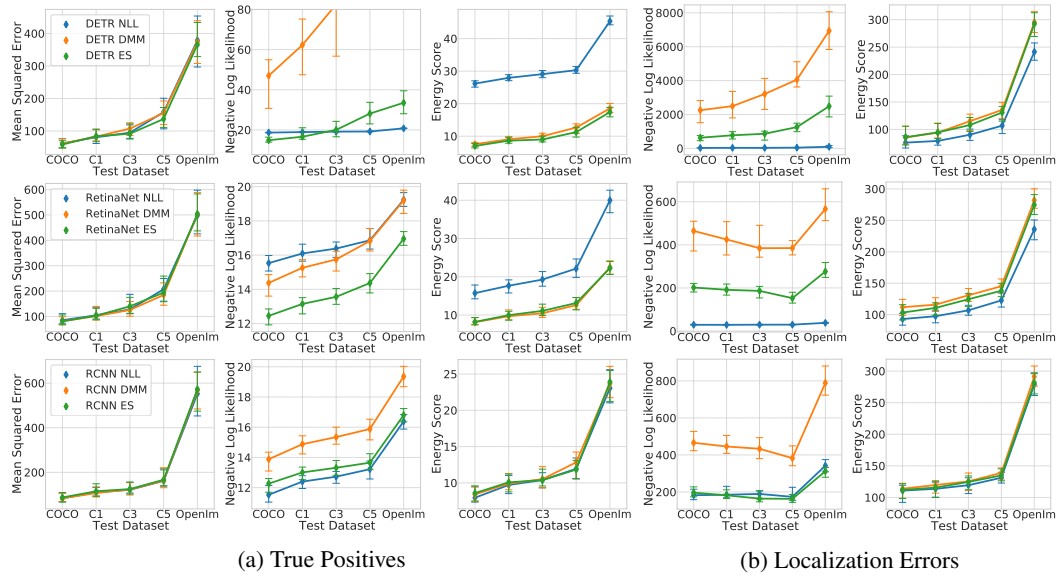

(a) True Positives       (b) Localization Errors

Figure 2: Average over $80$ classification categories of NLL, ES, and MSE for bounding box predictive distributions estimates from probabilistic detectors with DETR, RetinaNet, and FasterRCNN backends on in-distribution (COCO), artificially shifted (C1-C5), and naturally shifted (OpenIm) datasets. Error bars represent the $95\%$ confidence intervals around the mean.

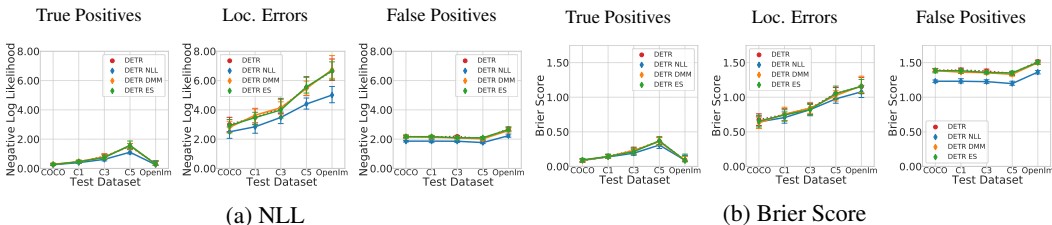

(a) NLL       (b) Brier Score

Figure 3: Average over $80$ classification categories of NLL and Brier score for classification predictive distributions generated using DETR. Error bars represent the $95\%$ confidence intervals around the mean. Similar trends are seen for RetinaNet and FasterRCNN backends in Figures F.2, F.3.

instances with $0.1 < IOU < 0.5$. Detections with $0.5 \leq IOU$ are considered true positives. If multiple detections have $0.5 \leq IOU$ with the same ground truth object, the one with the lower classification score is considered a duplicate. In practice, we report the average of all scores for true positives and duplicates at multiple IOU thresholds between $0.5$ and $0.95$, similar to how mAP is evaluated for the COCO dataset. Table 1 shows that the number of output objects in each of the four partitions is very similar for probabilistic detectors sharing the same backend, reinforcing the fairness of our evaluation. Duplicates are seen to comprise a small fraction of the output detections, and analyzing them is not found to provide any additional insight over evaluation of the other three partitions (See Figure F.1). Partitioning details and IOU thresholds can be found in Appendix C.

True positives and localization errors have corresponding ground truth targets, their predictive distributions can be evaluated using proper scoring rules. As non-local rules, we use the *Brier score* (Brier, 1950) for evaluating categorical predictive distributions and the *energy score* for evaluating bounding box predictive distributions. As a local rule, we use the *negative log likelihood* for evaluating both. False positives are not assigned a ground truth target; we argue that these output instances should be classified as background and assign them the background category as their classification ground truth target. The quality of false positive categorical predictive distributions can then be evaluated using NLL or the Brier score. Bounding box targets cannot be assigned to false positives and as such we only look at their differential predictive entropy. In addition to using proper

Table 1: **Left**: Results of mAP and calibration errors of probabilistic extensions of DETR, RetinaNet, and FasterRCNN. **Right**: The number of output detection instances classified as true positives (TP), duplicates, localization errors, and false positives (FP).

| Detector | Loss | mAP(%) ↑ | | MCE (Cls) ↓ | | CE (Reg) ↓ | | Counts Per Partition (in-dist) | | | |
|---|---|---|---|---|---|---|---|---|---|---|---|
| | | COCO | OpenIm | COCO | OpenIm | COCO | OpenIm | TP | Dup. | Loc. Errors | FP |
| DETR | NLL | 37.10 | 35.16 | 0.0735 | **0.0973** | 0.0512 | 0.0331 | 24452 | 1103 | 11228 | 12604 |
| | DMM | 39.82 | 38.88 | 0.0737 | 0.0996 | 0.0091 | 0.0171 | 24929 | 1197 | 10374 | 10406 |
| | ES | **39.96** | **39.31** | **0.0728** | 0.0993 | **0.0068** | **0.0167** | 24914 | 1216 | 10236 | 9761 |
| RetinaNet | NLL | 35.11 | 34.60 | 0.0260 | 0.0385 | 0.0242 | 0.0176 | 22733 | 2835 | 8992 | 6654 |
| | DMM | **36.28** | 34.84 | **0.0244** | 0.0372 | 0.0102 | 0.0189 | 23096 | 2783 | 8902 | 6411 |
| | ES | 36.02 | **34.97** | 0.0245 | 0.0370 | **0.0059** | **0.0138** | 23096 | 2783 | 8902 | 6411 |
| FasterRCNN | NLL | 37.14 | 34.30 | **0.0490** | **0.0713** | 0.0049 | 0.0133 | 23145 | 4841 | 7323 | 6989 |
| | DMM | 37.24 | 34.42 | 0.0496 | 0.0854 | 0.0112 | 0.0229 | 23187 | 4628 | 7165 | 6905 |
| | ES | **37.32** | **34.75** | 0.0495 | 0.0849 | 0.0051 | **0.0121** | 23123 | 4612 | 7208 | 6760 |

scoring rules, we provide mAP, classification marginal calibration error (MCE) (Kumar et al., 2019) and regression calibration error (Kuleshov et al., 2018) results for all methods in Table 1.

## 4.2 RESULTS ANALYSIS

Figures 2 and 3 show the results of evaluating the classification and regression predictive distributions for true positives, localization errors, and false positives under dataset shift and using proper scoring rules. Figure 2 also shows that the bounding box mean squared errors (MSE) for probabilistic extensions of the same backend are very similar, meaning that differences between regression proper scoring rules among these methods arise from different predictive covariance estimates and not predictive mean estimates. Similar to what has been reported by Ovadia et al. (2019) for pure classification tasks, we observe that the quality of both category and bounding box predictive distributions for all output partitions degrades under dataset shift. Probabilistic detectors sharing the same detection backend are shown in Figure 3 to have similar classification scores, which we expect given that we do not modify the classification loss function. However, when comparing regression proper scoring rules in Figure 2, the rank of methods can vary based on which proper scoring rule one looks at, a phenomenon we explore later in this section.

**Advantages of Our Evaluation:** Independently evaluating the quality of predictive distributions for various types of errors leads to a more insightful analysis when compared to standard evaluation using mAP or PDQ (Hall et al., 2020)[3]. As an example, Figure 3 shows probabilistic extensions of DETR to have a lower classification NLL, but a higher Brier score for their false positives when compared to their localization errors. We conclude that for its false positives, DETR assigns high probability mass to the correct target category while simultaneously assigning high probability mass to a small number of other erroneous categories, leading to lower NLL but a higher Brier score when compared to localization errors for which the probability mass is distributed across many categories. Our observation highlights the importance of using non-local proper scoring rules alongside local scoring rules for evaluation.

**Pitfalls of Training and Evaluation Using NLL:** Figure 4 shows the differential entropy of bounding box predictive distributions plotted against the error, measured as the IOU of their means with ground truth boxes. When using DETR or RetinaNet as a backend, variance networks trained with NLL are shown to predict higher entropy values when compared to those trained with ES regardless of the error. This observation does not extend to the FasterRCNN backend, where variance networks trained with NLL and ES are seen to have similar predictive entropy values. Figure 4 shows the distribution of errors, measured as IOU with ground truth, of targets used to compute regression loss functions during training of all three detection backends. At all stages of training, DETR and RetinaNet backends compute regression losses on targets with a much lower IOU than FasterRCNN, which are seen in Figure 4 as low IOU tails in their estimated histograms. *We observe a direct correlation between the number of low IOU regression targets used during training and the overall magnitude of the entropy of predictive distributions learnt using NLL.* DETR, using

---

[3]For more details see Appendix D.3

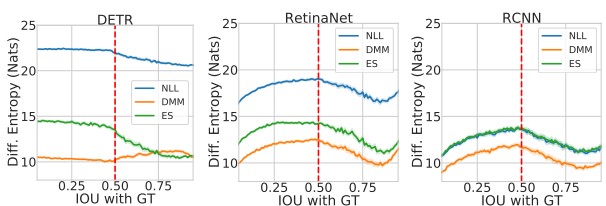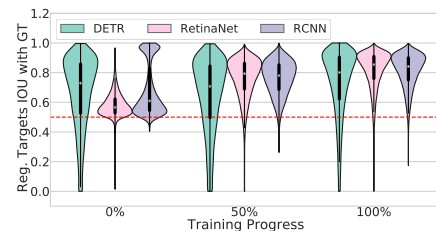

Figure 4: **Left**: Differential Entropy vs IOU with ground truth plots for bounding box predictive distribution estimates on in-distribution data. **Right**: Histograms of the IOU of ground truth boxes with boxes assigned as regression targets during network training, plotted at $0\%$, $50\%$, and a $100\%$ of the training process. The red dashed line signifies the $0.5$ IOU level on both plots.

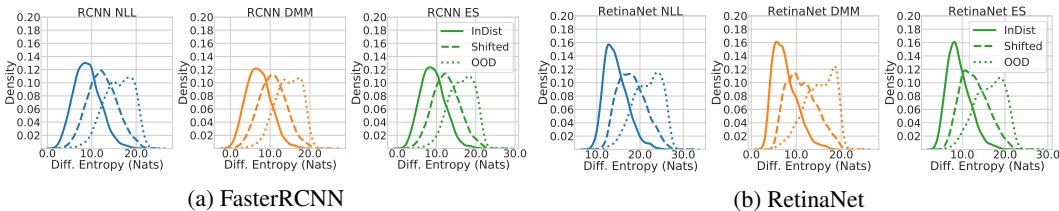

Figure 5: Histogram of differential entropy for **false positives** bounding box predictive distributions produced by probabilistic detectors with FasterRCNN and RetinaNet as a backend. Results for DETR exhibit similar trends (Figure F.4).

the largest number of low IOU regression targets is seen in Figure 4 to learn the highest entropy predictive distributions, while FasterRCNN using the fewest number of low IOU regression targets learns the lowest entropy predictive distributions. Variance networks trained with ES do not exhibit similar behavior producing a consistent magnitude for entropy regardless of their backend. Table 1 shows NLL to have a $\sim 2.7\%$, $\sim 0.9\%$, and $\sim 0.14\%$ reduction in mAP compared to ES and DMM when used to train DETR, RetinaNet, and FasterRCNN, respectively. Figure 4 shows that the drop in mAP when using NLL for training is directly correlated to the number of low IOU regression targets chosen during training by the three deterministic backends. Table 1 also shows low entropy distributions of DETR-ES and RetinaNet-ES to achieve a much lower regression calibration error than high entropy distributions of DETR-NLL and RetinaNet-NLL.

Columns 2 and 3 of Figure 2 show DETR-ES and RetinaNet-ES to have lower negative log likelihood and energy score for bounding box predictive distributions of true positives when compared to DETR-NLL and RetinaNet-NLL. For localization errors (Columns 4 and 5), the deviation from the mean is large, and as such the high predictive entropy provided by DETR-NLL and RetinaNet-NLL leads to lower values on proper scoring rules when compared to DETR-ES and RetinaNet-ES. We notice that if one uses only NLL for evaluation, networks can achieve a constant value of NLL by estimating high entropy predictive distributions regardless of the deviation from the mean, as seen for DETR-NLL on true positives in Figure 2. We can mitigate this issue by evaluating with the energy score, which is seen to distinguish between correct and incorrect high entropy distributions. As an example, DETR-NLL is shown to have lower negative log likelihood but a much higher Energy score when compared to DETR-ES for true positives on shifted data. On the other hand, DETR-NLL shows a slightly lower energy score when compared to DETR-ES on localization errors, meaning the energy score can indicate that the high entropy values provided by DETR-NLL are a better estimate for localization errors than the low entropy ones provided by DETR-ES. Considering that true positives outnumber localization errors by at least two in Table 1, we argue that it is more beneficial to train with the energy score for higher quality true positives predictive distributions over training with the negative log likelihood and predicting high entropy distributions regardless of the true error.

Evaluating only with the energy score also has disadvantages, we show that it does not sufficiently discriminate between the quality of low entropy distributions. Figure 2 shows that DMM-trained networks predicting the lowest entropy (Figure 4), achieve similar values of the energy score, but

much higher values of negative log likelihood when compared to networks trained with ES. The higher NLL score seen in Figure 2 for all networks trained with DMM a lower quality distributions when compared to networks trained with ES, specifically at the correct ground truth target value.

**Pitfalls of Common Approaches For Regression Target Assignment:** Figure 4 shows the entropy for all methods with the DETR backend to steadily decrease as a function of decreasing error. On the other hand, the entropy of methods using RetinaNet and FasterRCNN backend is seen to have two inflection points, one at an IOU of $0.5$ and another at a higher IOU of around $0.9$. As a function of decreasing error, the entropy increases before the first inflection point, decreases between the two, and then increases again after the second inflection point. *We hypothesize that this phenomenon is caused by the way backends choose their regression targets during training.* DETR uses optimal assignment to choose regression targets that span the whole range of possible IOU with GT, even during final stages of training, as is visible in Figure 4. On the other hand, RetinaNet and FasterRCNN use ad-hoc assignment with IOU thresholds Ren et al. (2015), a method that is seen to provide regression targets concentrated in the $0.5$ to $0.9$ IOU range throughout the training process, resulting in a much narrower data support when compared to DETR. We conclude that outside the data support, variance networks with RetinaNet and FasterRCNN backends fail to provide uncertainty estimates that capture the quality of mean predictions. Our conclusion is not unprecedented, Detlefsen et al. (2019) has previously shown variance networks to perform poorly out of the training data support for multiple regression tasks. However, our analysis pinpoints the reason of such behavior in probabilistic object detectors, showing that well established training approaches based on choosing high IOU regression targets work well for predictive mean estimation, but are not necessarily optimal for estimating predictive uncertainty.

**Performance on OOD Data:** Finally, Figure 5 shows histograms of regression predictive entropy of false positives from probabilitic detectors with DETR and FasterRCNN backend on in-distribution, naturally shifted, and out-of-distribution data. Variance networks trained with any of the three loss functions considered achieve the highest predictive differential entropy on out-of-distribution data, followed by the skewed data and achieve the lowest entropy on the in-distribution data, showing that variance networks are capable of reliably capturing dataset shift when used to predict uncertainty.

## 5 TAKEAWAYS

We propose to use the energy score, a proper and non-local scoring rule to train probabilistic detectors. Answering the call to aim for more reliable benchmarks in numerous setups of uncertainty estimation (Ashukha et al., 2020) we also present tools to evaluate probabilistic object detectors using well-established proper scoring rules. We summarize our main findings below:

- No single proper scoring rule can capture all the desirable properties of category classification and bounding box regression predictive distributions in probabilistic object detection. We recommend using both local and non-local proper scoring rules on multiple output partitions for a more expressive evaluation of probabilistic object detectors.

- Using a proper scoring rule as a minimization objective does not guarantee good predictive uncertainty estimates for probabilistic object detectors. Non-local rules, like the energy score, learn better calibrated, lower entropy, and higher quality predictive distributions when compared to local scoring rules like NLL.

- IOU-based assignment approaches used by established object detectors for choosing regression targets during training overly restrict data support to high IOU candidates, leading to unreliable bounding box predictive uncertainty estimates over the full range of possible errors.

- Variance networks are capable of reliably outputting increased predictive differential entropy when presented with out-of-distribution data for the bounding box regression task.

In this paper, we provide better tools for estimation and assessment of predictive distributions, by using variance networks with *existing object detection architectures* as a backend. Designing novel architectures that aim for accurate bounding box predictive distributions, rather than just accurate predictive means, remains an important open question.

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

# A EXPERIMENTAL DETAILS

## A.1 MODEL IMPLEMENTATION

The implementation of DETR [4], RetinaNet, and FasterRCNN models is based on original PyTorch implementations available under the *Detectron2* (Wu et al., 2019) object detection framework. RetinaNet, FasterRCNN, and DETR do not directly estimate bounding box corners, but work on a transformed bounding box representation $\mathbf{b} = T(\mathbf{z})$ where $T : \mathbb{R}^4 \to \mathbb{R}^4$ is an invertible transformation with inverse $T^{-1}(.)$. As an example, FasterRCNN estimates the difference between proposals generated from an region proposal network and ground truth bounding boxes. We train all probabilistic object detectors to estimate the covariance matrix $\mathbf{\Sigma}_b$ of the transformed bounding box representation $\mathbf{b}$. To predict positive semi-definite covariance matrices, each of these three object detection architectures are extended with a covariance regression head that outputs the 10 parameters of the lower triangular matrix $\mathbf{L}$ of the Cholesky decomposition $\mathbf{\Sigma}_b = \mathbf{L}\mathbf{L}^\mathsf{T}$. The diagonal parameters of the matrix $\mathbf{L}$ are passed through the exponential function to guarantee that the output covariance $\mathbf{\Sigma}_b$ is positive semi-definite. We train two models for each of the 9 architecture/regression loss combination, one using a full covariance assumption and the other using a diagonal covariance assumption. We report the results of the top performing variant chosen according to its performance on mAP as well as regression proper scoring rules. Considering the covariance structure as a hyperparameter was necessary for fair evaluation as we could not get RetinaNet to converge with NLL and a full covariance assumption (See Figure F.5). In case of a diagonal covariance matrix assumption, off-diagonal elements of $\mathbf{\Sigma}_b$ are set to 0. The covariance prediction head for each object detection architecture is an exact copy of the bounding box regression head used by that architecture, taking the same feature maps as input. Variance networks are initialized to produce identity covariance matrices, which we find to be key for stable training, especially for NLL loss.

## A.2 MODEL INFERENCE

When evaluating our probabilistic detectors one needs to be agnostic to the source of probabilistic bounding box predictions. As such, all considered methods are required to provide a consistent output bounding box representation and a corresponding covariance matrix $\mathbf{\Sigma}$ as presented in Section 3. We approximate $\mathbf{z} \sim \mathcal{N}(\boldsymbol{\mu}, \mathbf{\Sigma})$ by drawing 1000 samples from $\mathbf{b} \sim \mathcal{N}(\boldsymbol{\mu}_b, \mathbf{\Sigma}_b)$, passing those through $T^{-1}(.)$, and then estimate $\boldsymbol{\mu}$ and $\mathbf{\Sigma}$ as the sample mean and covariance matrix. **Note that because of $T(.)$ a diagonal $\mathbf{\Sigma}_b$ does not in general lead to a diagonal $\mathbf{\Sigma}$.** Other than the special consideration required to estimate the final bounding box probability distribution, the inference process is fixed to be the one provided in the original implementation for all detectors.

## A.3 MODEL TRAINING

For training all probabilistic extensions of the three object detectors, We stick to the hyperparameters provided by the original implementation whenever possible. Complex architectures such as DETR need a couple of weeks of training on our hardware setup and as such searching for the optimal values of hyperparameters for each tested configuration is outside the scope of this paper. All models are trained using a fixed random seed, which is shared across all APIs (numpy, torch, detectron2, etc...). This insures that empirical results are not determined based on lucky convergence, we train using 5 random seeds per configuration and find the results to be consistent with very small variance in terms of mAP and probabilistic metrics.

**RetinaNet and FasterRCNN** both use ResNet-50 followed by a feature pyramid network (FPN) for feature extraction. Since both models are trained with SGD and momentum in the original implementation, we use the linear scaling rule to scale down from batch size of 16 to a batch size of 4 due to hardware limitations. We train our probabilistic extensions of those models using 2 GPUS with a learning rate of 0.0025 for RetinaNet and 0.005 for FasterRCNN. Both models are trained for 270000 iterations, and the learning rate is dropped by a factor of 10 at 210000 and then again at 250000 iterations. All additional hyperparameters are left intact. Both RetinaNet and Faster-RCNN was trained using a soft mean warmup stage (Detlefsen et al., 2019). This is achieved through loss

---

[4]https://github.com/facebookresearch/detr/tree/master/d2

annealing, where the bounding box regression loss is defined as:

$$L_{reg} = (1 - \lambda)L_{original} + \lambda L_{probabilistic},$$
$$\omega = \min(1, \frac{i}{250000})$$
$$\lambda = \frac{100^{\omega} - 1}{100 - 1}, \tag{5}$$

where $L_{original}$ is the regression loss in the original non-probabilistic implementation of the respective object detector, and $i$ is the current training step. The value of $100$ as a base for the exponent was chosen using hyperparameter tuning. This loss formulation ensures that the network starts by emphasizing learning of the bounding box, slowly shifting to learning the probabilistic regression loss as training proceeds. For the last 20000 steps, only the probabilistic regression loss is used for training. We found this loss formulation to be essential for convergence of models trained using the NLL. Each FasterRCNN model takes $\sim 3$ days to train using 2 P-100 GPUs. On the same setup, RetinaNet models take $\sim 4$ days to finish training.

**DETR** also uses ResNet-50 as a base feature extractor. DETR's original implementation requires a very long training schedule for convergence, leading us to use hard mean warmup for all probabilistic DETR models. We use the model parameters provided by DETR's authors after training for 500 epochs to reach 42% mAP on the COCO validation dataset. Weights from this deterministic model are used as initial weights of all probabilistic extension of DETR, which are trained for an additional 50 epochs using losses presented in Section 3 and the same hyperparameters of the original deterministic implementation. We reduce the batch size from 64 to 16, but use the same initial learning rate as DETR is trained with ADAM. The learning rate is then dropped by a factor of 10 after at 30 epochs. Training for 50 epochs takes $\sim 4$ days using 4 T-4 GPUs.

# B    SHIFTED AND OOD DATASETS

## B.1    SHIFTING COCO DATASET WITH IMAGENET-C CORRUPTIONS

We corrupt the COCO validation dataset using 18 corruption types proposed in ImageNet-C (Hendrycks & Dietterich, 2019) at 5 increasing levels of intensity. The frames of COCO validation dataset are skewed using every corruption type in repeating sequential order, such that the first corruption type is applied to frame $1, 19, 37, \ldots$, the second to frames $2, 20, 38, \ldots$, and so on. By increasing the corruption intensity from level 1 to level 5, we create 5 shifted versions of the 5000 frames of the COCO validation dataset such that every frame is skewed with the same corruption type, but at an increasing intensity.

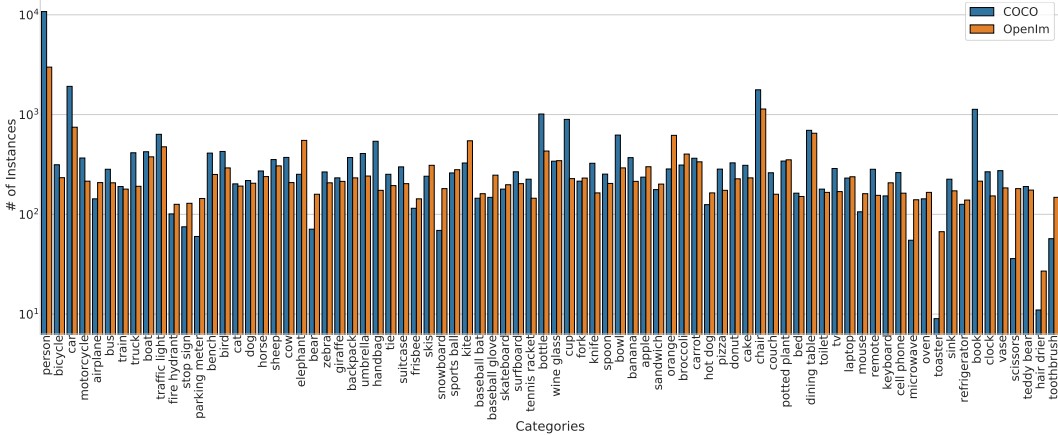

Figure B.1: Histogram comparing the number of instances of every category present in the COCO validation dataset with those present in our shifted dataset constructed from OpenImages-V4 frames.

### B.2 SHIFTED AND OUT OF DISTRIBUTION DATASETS FROM OPENIMAGES-V4

**Shifted Dataset**: To test methods beyond artificially shifted datasets, we create a new dataset comprising of $9,351$ frames from OpenImages-V4 (Kuznetsova et al., 2020) 2D detection data, containing instances belonging to the 80 object categories found in the COCO dataset. This testing data is naturally shifted due to differences in image quality, and different labeling mechanisms employed to generate the ground truth object boxes. Figure B.1 shows the number of instances belonging to each one of the 80 categories in the COCO validation dataset when compared to our generated OpenImages dataset. We tried to maintain the balance of categories found in the COCO validation dataset, as we aim to highlight performance differences originating from distribution shift, rather than the number of instances per category.

**Out-Of-Distribution Dataset**: To test methods on out-of-distribution data, we collect $1,852$ frames from OpenImages-V4, containing no instances from any of the 80 categories found in the COCO dataset. All frames were manually checked to minimize the existence of unlabeled in-distribution category instances.

## C PARTITIONING ERRORS IN OBJECT DETECTION

To maximize mAP, all our detection architectures are designed to produce a fixed number of detections per frame, usually a 100 (Carion et al., 2020), regardless of their classification score. To avoid performing our probabilistic evaluation on objects that can be trivially eliminated by a score threshold, we filter the output of our probabilistic detectors based on a classification score that maximizes the F-1 score on the COCO in-distribution dataset.

To analyze the predictive distributions provided by object detection models, we partition their filtered results into mutually exclusive subsets by adapting the object detection error decomposition presented by Hoiem et al. (2012). For every detection instance, the intersection-over-union(IOU) is calculated with every ground truth instance in the scene to determine the largest IOU, $\text{IOU}_{max}$. Based on $\text{IOU}_{max}$, detection instances are partitioned into false positives, true positives, localization errors, or duplicates. **False positives** are determined as detection instances for which $\text{IOU}_{max} \leq 0.1$ whereas **localization errors** are defined as ones with $0.1 < \text{IOU}_{max} < 0.5$. Multiple localization errors can be assigned the same ground truth detection; we argue that such duplication is an artifact of failed post-processing stages usually found in modern object detectors (non-maximum suppression for instance) and should neither be ignored nor lumped with false positives. To avoid a discussion on the definition of **true positives**, we define an IOU threshold $\eta_{tp} \in \{0.5, 0.55, \ldots, 0.95\}$ and consider the detection with $IOU_{max} \geq \eta_{tp}$ a true positive detection. For a finer error decomposition, if two detections are assigned the same ground truth instance, the one with a lower classification score is considered a **duplicate**. The choice of $\eta_{tp}$ is inspired by how mean average precision is computed on the COCO (Lin et al., 2014). *By computing averages of evaluation metrics on true positives and*

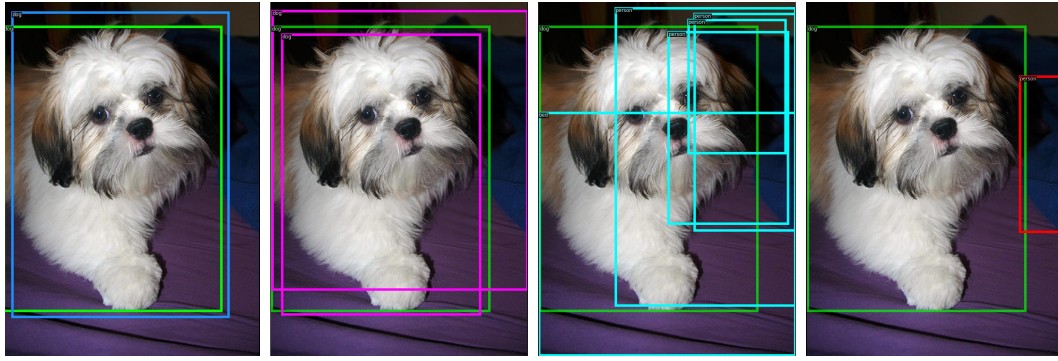

Figure C.1: Detection instances from deterministic Faster-RCNN partitioned into true positives (**blue**), duplicates (**magenta**), localization errors (**teal**), and false positives (**red**) according to $\eta_{tp} = 0.5$.

*duplicates determined using multiple IOU thresholds, we aim for a fair evaluation for models that are skewed for better performance at either a low or high IOU.*

Unlike the error decomposition presented in Hoiem et al. (2012), we do not combine duplicates with localization errors as they are expected to have different predictive uncertainty qualities. Furthermore, we do not put any constraint on the category of ground truth instances assigned to predicted objects. We argue that well localized but miss-classified object instances should not be counted as false positives, but should instead have their predictive categorical distribution evaluated through classification proper scoring rules (Ovadia et al., 2019). Finally, we do not consider **false negatives** in our evaluation as analyzing this type of error from a probabilistic detection prospective does not provide additional insights beyond what was provided by Hoiem et al. (2012).

## D    EVALUATING PREDICTIVE DISTRIBUTIONS

With the increase in interest surrounding the estimation of predictive distributions in recent deep learning literature, correctly evaluating such distributions given members of a test dataset is of utmost importance. This section aims to explain what characteristics we would like to see in high quality predictive distributions. It also provides a mathematical explanation of proper scoring rules.

### D.1    CALIBRATION AND SHARPNESS

Given a predictive distribution $p(\mathbf{z}|\mathbf{x}, \mathcal{D}; \boldsymbol{\theta})$ learnt using a training dataset $\mathcal{D}$, and a testing dataset $\mathcal{D}' = \{\boldsymbol{x}'_n, \boldsymbol{z}'_n\} \mid n \in \{1, \dots, N'\}$, we need a scoring rule $\mathcal{S}$ that assigns a numerical score to the predictive distribution $p(\mathbf{z}|\boldsymbol{x}'_n, \mathcal{D}; \boldsymbol{\theta})$ given the actual event $\boldsymbol{z}'_n$ that materialized in the testing dataset. A question naturally arises on the nature of the qualities of the predictive distributions than need to be captured by $\mathcal{S}$. Gneiting & Raftery (2007) contended that the goal of a predictive distribution is to maximize the *sharpness* around the materialized event $\boldsymbol{z}'_n$ subject to *calibration*. Calibration (Kuleshov et al., 2018; Kumar et al., 2019) is a joint property of the estimated predictive distribution as well as the events or values that materialized that reflects their statistical consistency (Gneiting & Raftery, 2007). In simple words, if a well-calibrated predictive distribution assigns a $0.8$ probability to an event, the event should occur around $80\%$ of the time. From its definition, one can see that calibration by itself is not enough to guarantee useful predictive distributions. As an example, a predictive distribution

$$p(\mathbf{z}|\mathbf{x}, \mathcal{D}) = \begin{cases} 1 \text{ if } \mathbf{z} = \mathbb{E}[\mathbf{z}] \\ 0 \text{ otherwise} \end{cases}$$

is perfectly calibrated, but not very useful (unless one wants to always predict $\mathbb{E}[\mathbf{z}]$). Sharpness on the other hand quantifies the concentration of the predictive distribution around the true materialized event $\boldsymbol{z}'_n$ and is *a property of the predictive distribution only*. Good predictions need to be sharp, but ideal predictions should be both sharp and well-calibrated. We would like our scoring rules $\mathcal{S}$ to capture both the sharpness and the calibration of a predictive distribution.

### D.2    PROPER SCORING RULES

Let $\Omega$ be a general sample space, $\mathcal{A}$ be a $\sigma$-algebra of $\Omega$, and $\mathcal{P}$ be a convex class of probability measure on $(\Omega, \mathcal{A})$ such that $\{p, p^* \in \mathcal{P}\}$. A scoring rule $\mathcal{S} : \mathcal{P} \times \Omega \to \mathbb{R}$ is written as $\mathcal{S}(p(\mathbf{z}|\boldsymbol{x}'_n, \mathcal{D}; \boldsymbol{\theta}), \boldsymbol{z}'_n)$ and maps a predictive distribution and a materialized event to a scalar value. Throughout the rest of this dissertation, scoring rules will be negatively oriented, that is the lower the score the better the predictive distribution.

With some abuse of notation, given the real data generating distribution $p^*(\mathbf{z}|\boldsymbol{x}'_n)$, we write the expected score under $p^*$ as:

$$\mathcal{S}(p(\mathbf{z}|\boldsymbol{x}'_n, \mathcal{D}; \boldsymbol{\theta}), p^*(\mathbf{z}|\boldsymbol{x}'_n)) = \int S(p(\mathbf{z}|\boldsymbol{x}'_n, \mathcal{D}; \boldsymbol{\theta}), \boldsymbol{z}'_n) \, \mathrm{d} \, p^*(\boldsymbol{z}'_n|\boldsymbol{x}'_n). \tag{6}$$

A scoring rule is said to be *proper* relative to $\mathcal{P}$ if:

$$\mathcal{S}\left(p^*(\mathbf{z}|\boldsymbol{x}'_n), p^*(\mathbf{z}|\boldsymbol{x}'_n)\right) \le \mathcal{S}\left(p(\mathbf{z}|\boldsymbol{x}'_n, \mathcal{D}; \boldsymbol{\theta}), p^*(\mathbf{z}|\boldsymbol{x}'_n)\right) \; \forall \, p, p^* \in \mathcal{P}, \tag{7}$$

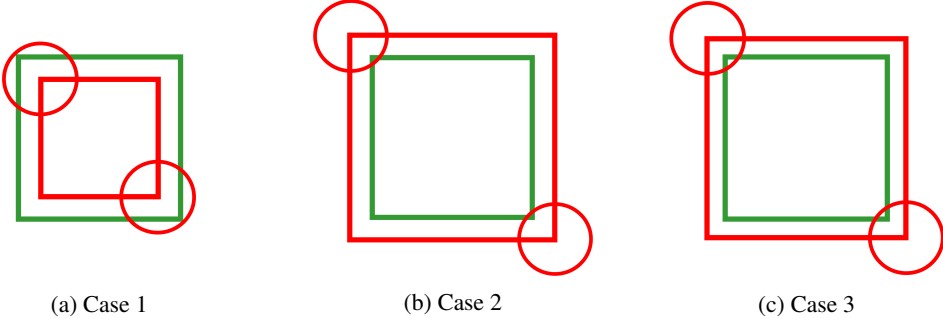

(a) Case 1      (b) Case 2      (c) Case 3

Figure D.1: A toy example showing the three bounding box distributions used to show the spatial quality $\mathcal{Q}_S$ to be a non-proper scoring rule. The green bounding box is the ground truth sample, whereas the red bounding box is the predicted mean of the probability distribution. The $95\%$ confidence ellipse of bounding box corner predictive distributions is also plotted in red.

Table 2: NLL, ES, and Spatial Quality ($\mathcal{Q}_S$) results of the three predictive distributions from our toy example.

| Case # | NLL $\downarrow$ | ES $\downarrow$ | $\mathcal{Q}_S \uparrow (\%)$ |
|--------|------|------|------|
| 1 | 20.49 | 23.10 | **51.88** |
| 2 | 20.49 | 23.10 | 45.98 |
| 3 | **19.33** | **21.21** | 50.02 |

and *strictly proper* relative to $\mathcal{P}$ if Equation equation 7 holds with equality *only if* $p = p*$. In simple words, a strictly proper scoring rule is only minimized if the predictive distribution is exactly equal to the true data generating distribution. In both cases, the a lower score signifies predictive distributions that are closer to the data generating distribution. It is clear that one can use proper scoring rules to rank predictive distributions based on theoretically founded quantities.

### D.3 PDQ IS NOT A PROPER SCORING RULE

PDQ can be written as:

$$\text{PDQ}(\mathcal{G}, \mathcal{D}) = \frac{1}{|\mathcal{G}| + N_{FP}} \sum_{i,j,f} pPDQ(\mathcal{G}_i^f, \mathcal{D}_j^f), \tag{8}$$

where $\mathcal{G}_i^f$ is the $i^{th}$ ground truth object instance in the $f^{th}$ frame of a dataset, $\mathcal{D}_j^f$ is the $j^{th}$ matched detection from the same frame, $|G|$ is the number of ground truth instances in the dataset, and $N_{FP}$ is the total number of false positives. The pPDQ can be written as:

$$\text{pPDQ}(\mathcal{G}_i^f, \mathcal{D}_j^f) = \sqrt{\mathcal{Q}_S(\mathcal{G}_i^f, \mathcal{D}_j^f) . \mathcal{Q}_L(\mathcal{G}_i^f, \mathcal{D}_j^f)}, \tag{9}$$

where $\mathcal{Q}_S$ is a spatial quality quantifying the quality of the bounding box predictive distribution, and $\mathcal{Q}_L$ is the label quality quantifying the quality of the categorical predictive distribution. Let us assume for the sake of our argument that both $\mathcal{Q}_S$ and $\mathcal{Q}_L$ are proper scoring rules. Different combinations of $\mathcal{Q}_S$ and $\mathcal{Q}_L$ can lead to the same value of pPDQ. An extreme example would be if $\mathcal{Q}_S = 0$, pPDQ will be 0 regardless of $\mathcal{Q}_L$. This would result in erroneous predictive distributions having the same of pPDQ as correct ones, meaning that PDQ cannot correctly rank probabilistic object detectors.

A more substantial problem with PDQ relates to the spatial quality $\mathcal{Q}_S$, which we empirically show to **not be a proper scoring rule**. To do so, we will generate a toy example consisting of one ground truth bounding box and three corresponding predictive probability distributions. The ground truth bounding box is defined using the top left and bottom right corners: $(u_{\min}, v_{\min})$, $(u_{\max}, v_{\max})$. We will refer to the first predictive distribution as **case 1**, and will assign it the following mean

$\mu_1 = (u_{\min} + 15, v_{\min} + 15)$, $(u_{\max} - 15, v_{\max} - 15)$, which is the ground truth bounding box shrunken by 15 pixels on the two image axes. The second predictive distribution, **case 2** will have $\mu_2 = (u_{\min} - 15, v_{\min} - 15)$, $(u_{\max} + 15, v_{\max} + 15)$, which is the ground truth bounding box expanded by 15 pixels on the two image axes. The final predictive distribution, **case 3**, will also be assigned the ground truth bounding box expanded with 14 pixels as: $\mu_3 = (u_{\min} - 14, v_{\min} - 14)$, $(u_{\max} + 14, v_{\max} + 14)$. All three predictive distributions will be assigned an identical $4 \times 4$ isotropic covariance matrix $\sigma^2 \boldsymbol{I}$ with $\sigma^2 = 50$. The three distributions, along with the ground truth bounding box, can be visually seen in Figure D.1.

Our argument is simple: if $\mathcal{Q}_S$ is a proper scoring rule, it should rank the three predictive distributions in an identical manner to NLL and ES. Table 2 shows the NLL, ES, and PDQ results of evaluating the three predictive distributions using the ground truth box sample. The first thing to note is that the NLL and ES values for cases 1 and 2 are identical. This is because both distributions share an identical covariance matrix, as well as an identical error with the ground truth bounding box (15 pixels in each image axis). However, the value of $\mathcal{Q}_S$ for case 1 is around $6\%$ higher than that of case 2. This phenomenon is due to the ad-hoc design choices used to define $\mathcal{Q}_S$ in Hall et al. (2020), particularly the way $\mathcal{Q}_S$ is built from a "foreground" and "background" components. A more definitive proof that $\mathcal{Q}_S$ is not proper stems from a comparison between the results of cases 1 and 3 in Table 2 (rows 1 and 3). Case 3 has an error of 14 pixels in every image axis when compared to the ground truth, $1 pixel$ less than the error of case 1. The lower error translates to a lower NLL, as well as ES values for case 3 when compared to case 1, implying that the predictive distribution of case 3 is of higher quality than that of case 1. However, when using $\mathcal{Q}_S$ for comparison between the two cases, case 1 has a $2\%$ better value when compared to case 3. In short, $\mathcal{Q}_S$ can provide a higher rank to lower quality predictive distributions when compared to higher quality ones (as measured by proper scoring rules such as NLL and ES), and as such is not a proper scoring rule.

In addition to not being a proper scoring rule, pPDQ is computed on a single output partition, constructed by using optimal assignment to match each ground truth object to a single detection output. This prevents in-depth analysis on the sources of errors such as the one presented in Section 4. Finally, PDQ has been previously shown to be gameable, where ad-hoc modifications of predictive distributions lead to performance gains, regardless of the theoretical soundness of such modifications. Contestants in a recent probabilistic object detection challenge (CVPR 2019) that relies on PDQ for evaluation showed that replacing the categorical output probability distribution with a one-hot vector representation led to strictly higher PDQ scores than any method attempting to accurately represent the data generating distribution (see Sections 4.3 in (Ammirato & Berg, 2019) and 3.2 in (Wang et al., 2019)). The gameability of PDQ highlights the importance of our suggestion to use proper scoring rules for evaluation, as practitioners are seen to be susceptible to letting go of theoretical soundness in favor of performance gains on evaluation metrics.

## E   MAXIMUM MEAN DISCREPANCY AND THE ENERGY DISTANCE

Maximum Mean Discrepancy (MMD) has been previously used to train generative models (Li et al., 2015; 2017), where minimizing MDD can be interpreted as matching the moments of the predicted model distribution to the empirical data distribution. In this work we are concerned with the Energy Distance (Rizzo & Székely, 2016), a maximum mean discrepancy (see Sejdinovic et al. (2013)) that is simple and efficient to estimate from distribution samples.

Given two independent random vectors $\mathbf{f}, \mathbf{g} \in \mathbb{R}^d$ with cumulative distribution function $F, G$ respectively, the squared Energy Distance (ED) can be written as:

$$\mathrm{D}^2(F, G) = 2\mathbb{E}||\mathbf{f} - \mathbf{g}|| - \mathbb{E}||\mathbf{f} - \mathbf{f}'|| - \mathbb{E}||\mathbf{g} - \mathbf{g}'||, \tag{10}$$

where $\mathbf{f}, \mathbf{f}'$ are i.i.d samples from $F$, and $\mathbf{g}, \mathbf{g}'$ i.i.d samples from $G$. Rizzo & Székely (2016) show that the energy distance satisfies all axioms of a metric, providing a measure of equality of distributions and insuring that $\mathrm{D}^2(F, G) = 0$ if and only if $F = G$. In context of deep learning, the energy distance has been previously used to parallelize text-to-speech generative models (Gritsenko et al., 2020), and to train generative adversarial networks (Bellemare et al., 2017). It can be shown (Gneiting & Raftery, 2007) that Equation 2 can be written as:

$$\mathrm{ES} = \mathbb{E}||\mathbf{f} - \mathbf{g}|| - \frac{1}{2}\mathbb{E}||\mathbf{f} - \mathbf{f}'|| \tag{11}$$

It is easy to see that the energy score presented in equation 11 is the energy distance when only one sample, $g$, is available from $G$.

# F  ADDITIONAL RESULTS

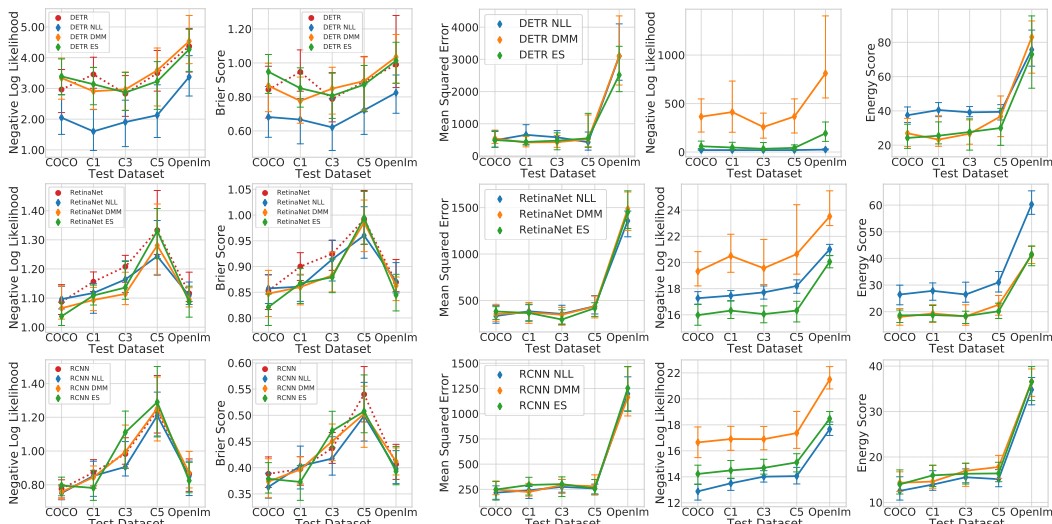

Figure F.1: Point plots showing the results of the mean squared errors, NLL, and Energy score when used to evaluate regression predictive distributions from from probabilistic detectors with DETR (Top), RetinaNet (Middle), and FasterRCNN (Bottom) backends on in-distribution (COCO), artificially shifted (C1-C5), and naturally shifted (OpenIm) datasets for **duplicate** detections. Results of regression metrics seem to have the same trend as those of true positives, while results of classification metrics follow those of localization errors. This is no surprise given true positives and duplicates can have the same IOU with ground truth, the only difference being duplicates having a lower class probability.

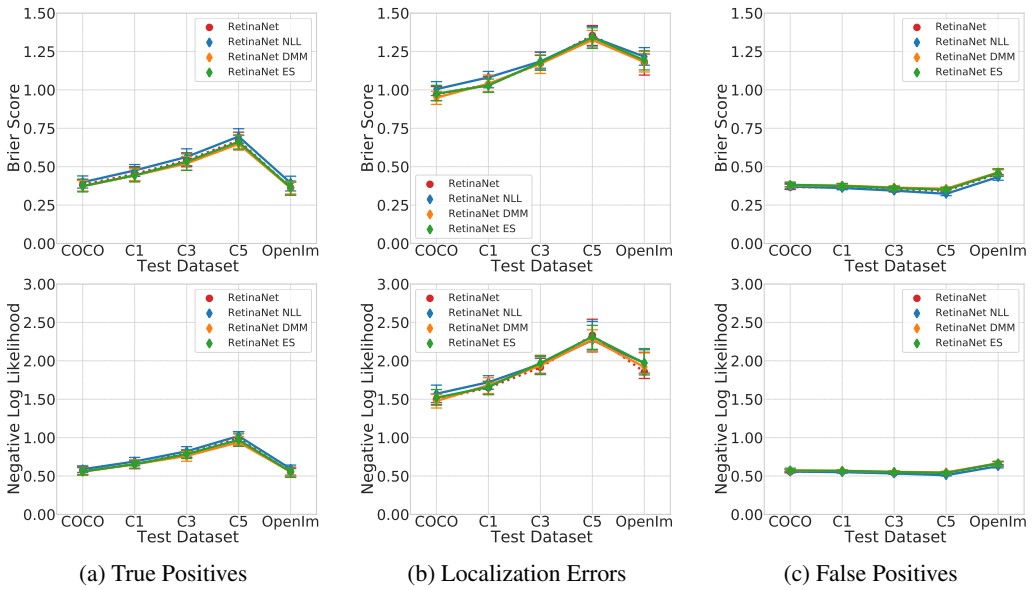

Figure F.2: Point plots of the Brier Score and NLL for results from RetinaNet. The same trends are seen as in Figure 3, with the expection of false positives having a lower Brier score than localization errors and duplicates. We suspect that this is due to usage of multilable classification with sigmoid instead of multiclass classification with softmax in the original RetinaNet implementation.

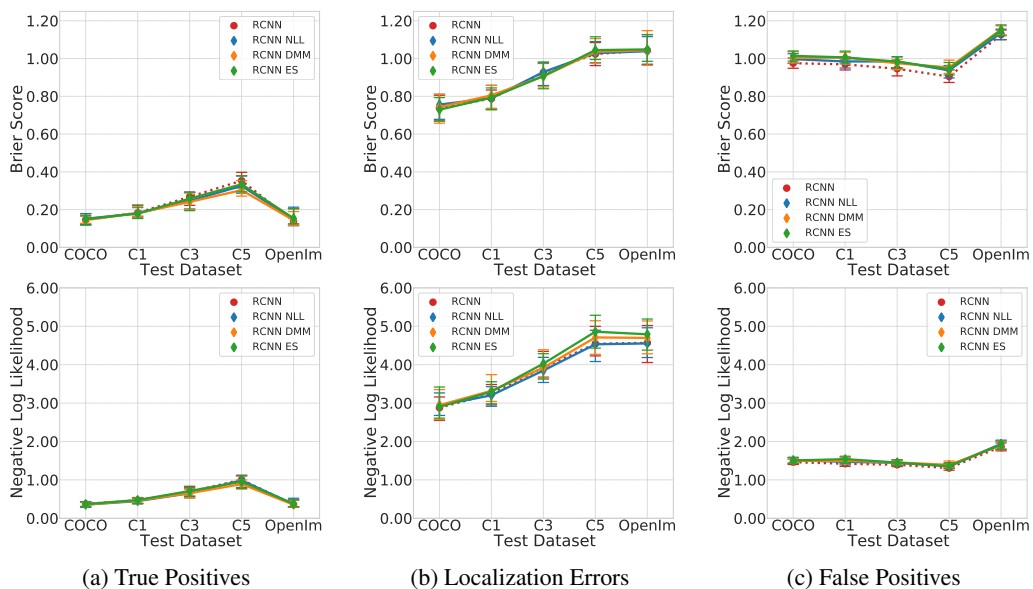

Figure F.3: Point plots of the Brier Score and NLL for results from FasterRCNN. The same trends are seen as in Figure 3.

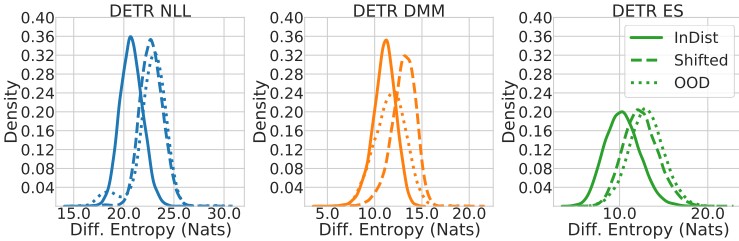

Figure F.4: Histogram of differential entropy for **false positives** bounding box predictive distributions on DETR. For networks trained using NLL and ES, the same trend is observed as in Figure 5. The network trained with DMM is shown to not be able to differentiate between in and out-of-distribution false positives.

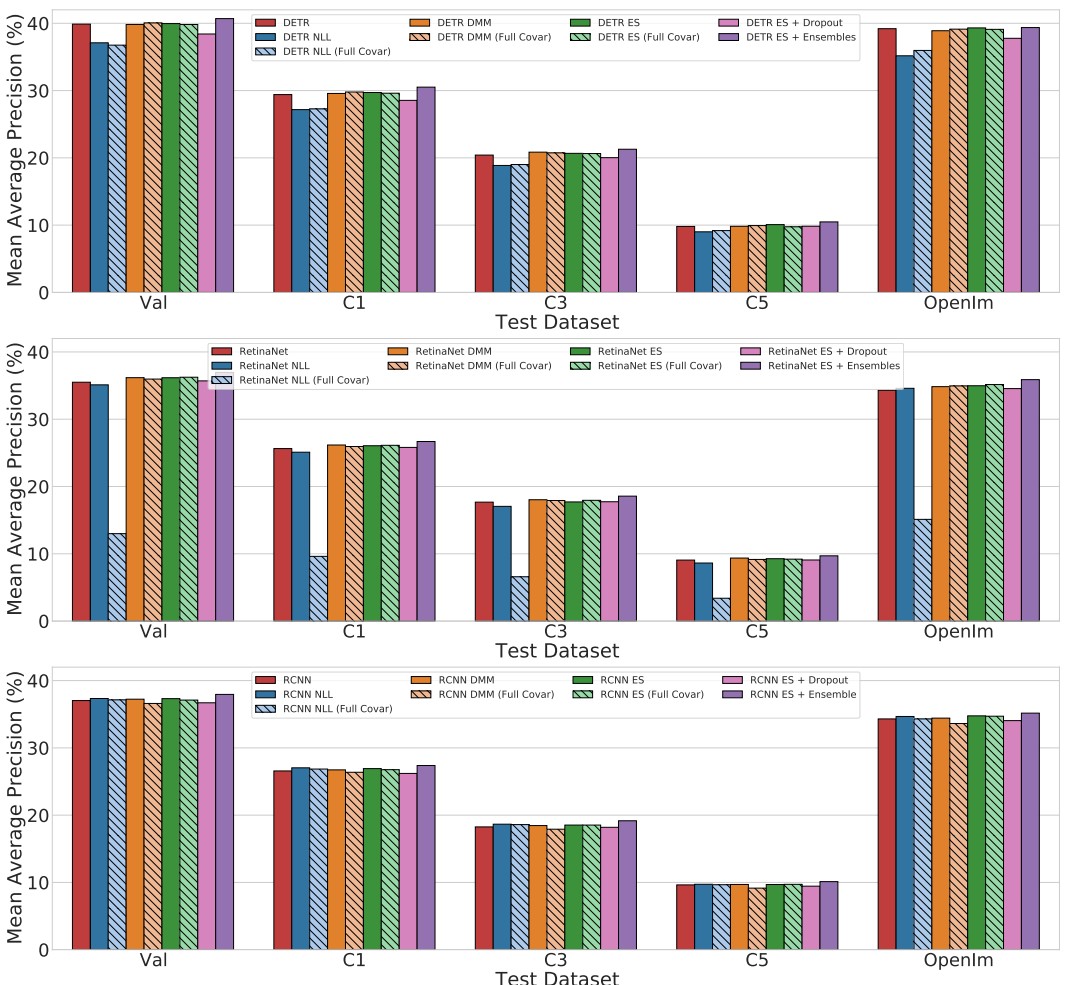

Figure F.5: Bar plots showing that our probabilistic detectors implementations achieve the same level of performance as their deterministic counterparts (in Red) on mean AP. On RetinaNet using NLL and a full covariance matrix assumption results in a model with much lower mAP than the original RetinaNet implementation.

# G  QUALITATIVE RESULTS

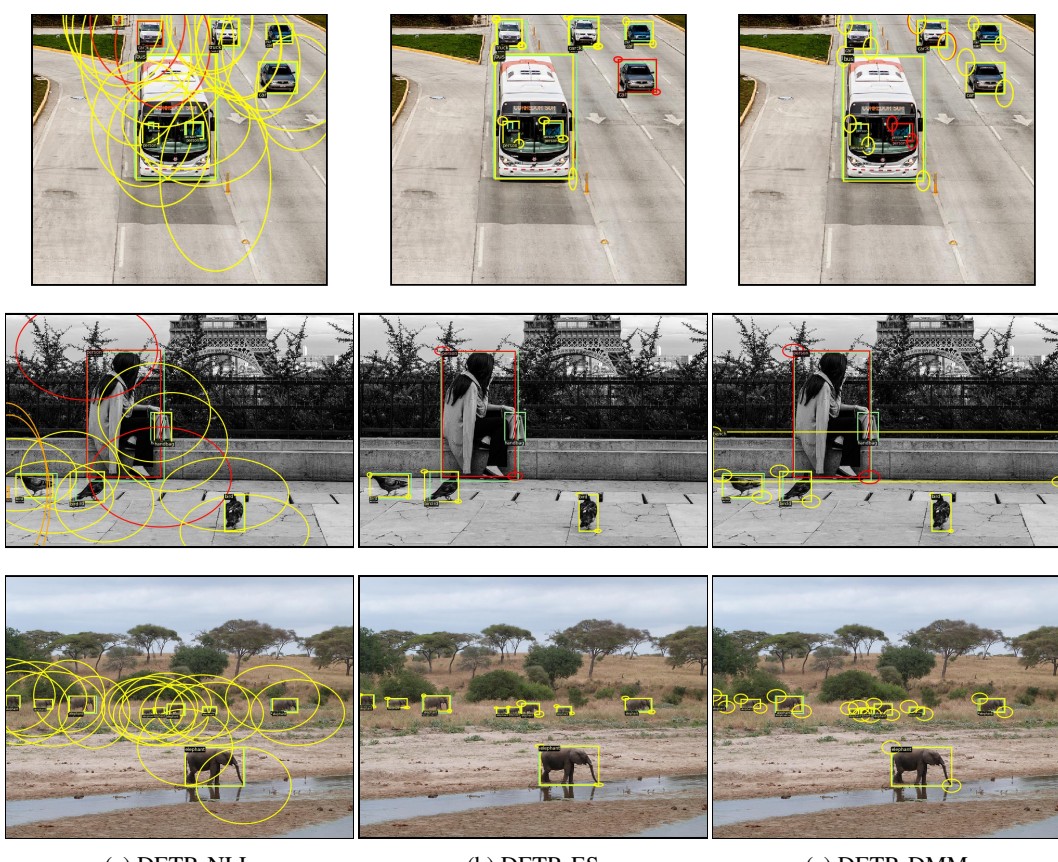

|        (a) DETR-NLL        |        (b) DETR-ES        |        (c) DETR-DMM        |

Figure G.1: Quantitative results from probabilistic detectors with a DETR backend. The color of bounding boxes signifies the entropy of their categorical distributions, where yellow means low entropy, orange means moderate entropy, and red means high entropy. We also plot the $95\%$ confidence ellipse of bounding box corner predictive distributions. The bounding box predictive distributions for DETR-NLL detections are of much higher uncertainty than the predictive distributions of DETR-ES and DETR-DMM detections. Ground truth bounding boxes are shown in green.

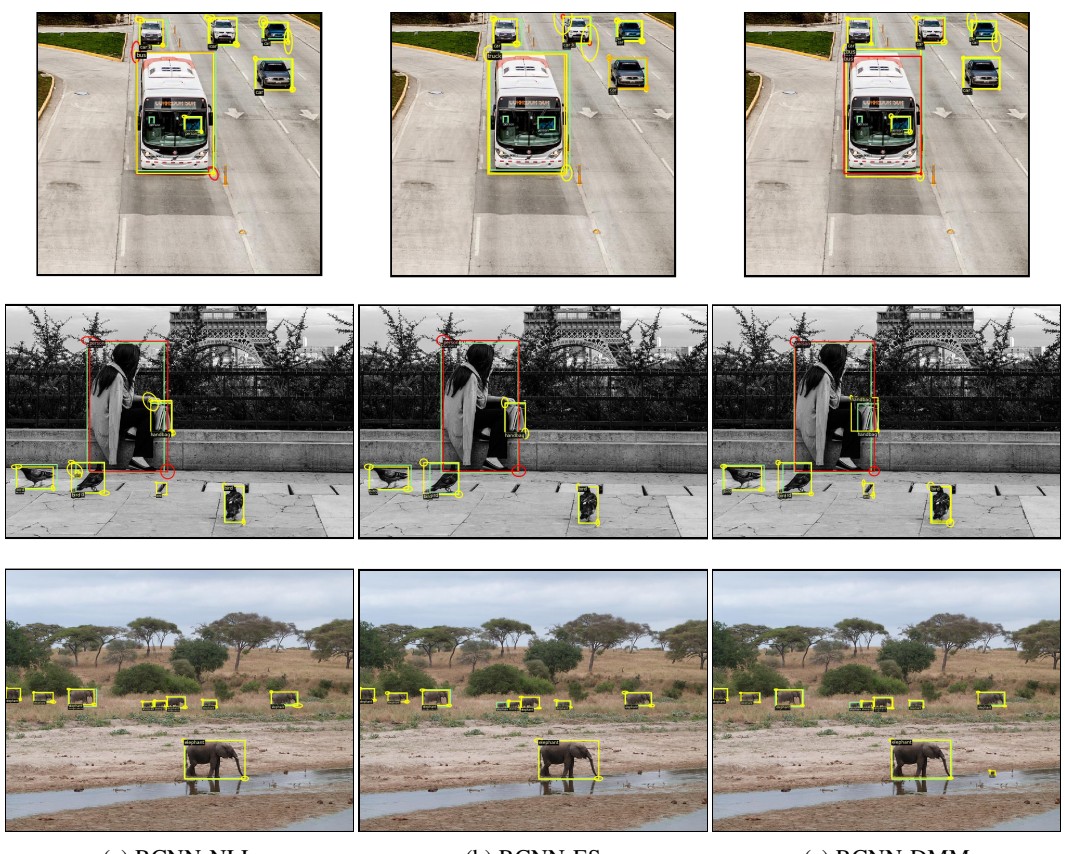

(a) RCNN-NLL       (b) RCNN-ES       (c) RCNN-DMM

Figure G.2: Quantitative results from probabilistic detectors with a Faster-RCNN backend. The color of bounding boxes signifies the entropy of their categorical distributions, where yellow means low entropy, orange means moderate entropy, and red means high entropy. We also plot the $95\%$ confidence ellipse of bounding box corner predictive distributions. It can be clearly seen that the bounding box predictive distributions for RCNN-NLL, RCNN-ES and RCNN-DMM have very similar uncertainty values. Ground truth bounding boxes are shown in green.

