# OpenReview forum: "Estimating and Evaluating Regression Predictive Uncertainty in Deep Object Detectors"
_ICLR.cc/2021/Conference — ICLR 2021 Poster_

### Official Review · AnonReviewer3 · 2020-10-27
**Exploring non-local proper scoring rule and assessing predictive distributions for detectors.**

**Rating:** 6
**Confidence:** 4

**Review:**

This paper introduces several ways to assess probabilistic object detectors that output predictive distributions. Moreover, it presents a proper and non-local scoring rule for training such probabilistic object detectors.  The model backbone consists of standard detectors (DETR, RetinaNet, FasterRCNN) with a variance regression head and a mean bounding box regression head.  The loss functions explored in the paper are the negative log likelihood (NLL, which is local and proper), energy score (ES, which is non-local and proper), and direct moment matching (DMM, which is non-local and non-proper), so this means that nine different combinations of detectors and losses can be formed.   The assessment uses NLL, MSE, ES and Brier Score on four partitions of the detection results (based on their IOU with ground truth): true positives, duplicates, localization errors, and false positives.  The training set is COCO, where there are three testing sets: the in-distribution test set is the COCO validation set, the artificially shifted data is the COCO validation dataset modified with 18 different image corruptions (Hendrycks & Dietterich, 2019), and the naturally shifted data is the OpenImages dataset (Kuznetsova et al., 2020).

The main results found by the paper are: 1) none of the proper score functions alone enables a complete analysis of the probabilistic object detectors -- for a thorough analysis, one should use both local and non-local proper scoring functions on multiple testing partitions; 2) proper scoring functions not necessarily provide good predictive uncertainty estimates for probabilistic object detectors, and non-local scoring functions allow better calibrated and lower entropy predictive distributions than local functions; 3) training based on IOU restricts results to high IOU candidates, which can lead to unreliable bounding box predictive uncertainty estimates; and 4) variance regression outputs large predictive differential entropy when presented with out-of-distribution data.

This paper has potential, but I found it not particularly well written.  In fact I found it hard to get its main point since it touches in many aspects of probabilistic object detectors, but none of the points are really clear. There are some other issues that need to be clarified -- please see below.
1- It is not clear from the formulation how the paper handles multiple detections
2- In Section 3.1, there are inconsistencies in the nomenclature -- x is written as boldface and then in italics, but they mean the same thing.
3- The experimental setup is not clear -- can you defined better "with any corresponding ground truth bounding box" (page 4)?
4- From the definitions of ES and DMM, it is not clear what is done with multiple instances of objects to be detected in an image.
5- Given the ES formulation, can we say that ES is prone to be overconfident?
6- The proofs for the scoring functions being proper or not should be more formal.
7- In Fig. 2, in each column the graphs should have the same range.
8- Letters in Table 1 are too small.
9- The definition of true positives, duplicates, localization errors, and false positives should be present.
10- On page 6, the "Advantages of our evaluation" sub-section only shows an anecdotal example of the advantages of the proposed measure.  It is expected a more direct comparison with competing approaches, such as PDQ.

After rebuttal, I raised my score from 4 to 6.

---

> ### Author Response · Authors · 2020-11-15
> **Authors' Reply To Reviewer 3**
>
> **This paper has potential, but I found it not particularly well written. In fact I found it hard to get its main point since it touches in many aspects of probabilistic object detectors, but none of the points are really clear.**
>
> We appreciate this feedback, and have made a concerted effort to make the paper more accessible to a wider audience. We have modified the introduction to better highlight the problem we try to solve as well as the contributions and insights. We have also modified Sections 3.4, 3.5, and Section 4 to better describe our experimental setup. We kindly ask the reviewer to point out any aspects of probabilistic object detectors that are still not clear in our updated write up, we will be happy to clarify any additional concerns.
>
> **It is not clear from the formulation how the paper handles multiple detections.**
>
> We would like to highlight that multiple object instances in the scene are handled in the same way as deterministic object detection backends (DETR, FasterRCNN, RetinaNet) handle them during both training and inference. For evaluation, metrics are computed for each output detection in the scene independently and then averaged to produce the final results. A more detailed explanation is presented in Appendix C of the supplemental materials. We add visualization showing multiple detection output from probabilistic extensions of DETR and FasterRCNN in Appendix G.
>
> **In Section 3.1, there are inconsistencies in the nomenclature -- x is written as boldface and then in italics, but they mean the same thing.**
>
> We would like to highlight we use boldface to represent random variables predicted by the network, and use italics to represent ground truth dataset elements. We added an explanation of this notation in Section 3.1.
>
> **The experimental setup is not clear -- can you defined better "with any corresponding ground truth bounding box" (page 4).**
>
> We agree with the reviewer that this statement is ambiguous, we modified the paper to make the statement clearer, as follows: "We split the detection results into a high error set with an IOU<=0.5, and a low error set with an IOU>0.5, where the IOU is determined as the maximum IOU with any ground truth bounding box in the scene".
>
> **From the definitions of ES and DMM, it is not clear what is done with multiple instances of objects to be detected in an image.**
>
> We would like to highlight that when using ES and DMM for training, multiple object instances are represented by $N$ in their respective equations. We would like to highlight that DMM uses a single prediction/ground truth bounding box pair per instance to try to match moments. The exact value of $N$ is determined by the regression target assignment scheme of the deterministic backends, usually taking into account object instances in all images of a training minibatch.
>
> **Given the ES formulation, can we say that ES is prone to be overconfident?**
>
> We agree with the reviewer that in theory ES could be prone to overconfidence as shown in Figure 1. However we have to note that our experiments with ES have shown no evidence for being overconfident when used for learning bounding box predictive distributions. In fact, looking at Figure 4, one can see that the entropy of distributions provided by NLL matches that of ES on FasterRCNN. This implies that when FasterRCNN is trained with NLL using high IOU regression targets, it tends to provide similar values of entropy to FasterRCNN trained with ES. On the other hand, with DETR and RetinaNet, NLL was underconfident while ES provided good values of entropy. This can be further confirmed by looking at the results on proper scoring rules in Figure 2, where ES is shown to outperform NLL on DETR and RetinaNet.
>
> **The proofs for the scoring functions being proper or not should be more formal.**
>
> We would like to highlight that NLL and ES have been formally proven to satisfy the definition of proper scoring rule in previous literature, we cite these papers in Sections 3.2-3.4. Proving mAP is not proper is trivial as it does not depend on the bounding box covariance estimates from probabilistic detectors. PDQ was shown to not be a proper scoring metric with an example in Appendix D. We have now added better explanations in Section 3.5 to give insights on why DMM is not proper.
>
> **In Fig. 2, in each column the graphs should have the same range.**
>
> We would like to highlight that Fig.2 is not meant to compare across architectures. We found that restricting the columns to have the same range will reduce the clarity of plots used to compare between NLL, ES, and DMM when used to train a specific deterministic backend.
>
> **Letters in Table 1 are too small.**
>
> We agree with the reviewer on this point, we increased the size of Table 1.

---

> > ### Author Response · Authors · 2020-11-15
> > **Cont'd**
> >
> > **The definition of true positives, duplicates, localization errors, and false positives should be present.**
> >
> > We agree with the reviewer on this point, and would like to mention that the definition of true positives, duplicates, localization errors, and false positives had a dedicated section in the appendix, Appendix C, and were not added to the original paper due to limited space. We modified the paper to add a summary of the definitions back to the experimental setup (Section 4.1, first paragraph).
> >
> > **On page 6, the "Advantages of our evaluation" sub-section only shows an anecdotal example of the advantages of the proposed measure. It is expected a more direct comparison with competing approaches, such as PDQ.**
> >
> > We would like to highlight that our evaluation provides a theoretically founded statistical evaluation of output predictive distributions, done independently for the classification and regression tasks as well as on four output partitions. As such, our evaluation cannot be directly compared to non-proper metrics such as PDQ which aggregate results for classification and regression over all outputs into a single scalar value.
> >
> > In addition, the PDQ evaluation metric has been shown to be gameable. Contestants in a 2019 CVPR probabilistic object detection challenge that relies on PDQ showed that replacing the categorical output probability distribution with a one-hot vector representation consistently resulted in the highest PDQ value, regardless of the correctness of the output predictive distribution (Section 4.3 in [1] and 3.2 in [2]). The gameability of PDQ highlights the importance of our submitted paper, as it shows the susceptibility of practitioners to let go of theoretical soundness in favor of performance gains on evaluation metrics. We avoided mentioning gameability issues in the paper as we are aware of PDQ's authors' attempts to fix these problems and would like to see them succeed in doing so. We added the above discussion to Appendix D, and referred to it in Section "Advantages of our evaluation".
> >
> > [1] Ammirato, Phil, and Alexander C. Berg. "A Mask-RCNN Baseline for Probabilistic Object Detection." CVPR workshop on the Robotic Vision Probabilistic Object Detection Challenge. 2019.
> >
> > [2] Wang, Chuan-Wei, et al. "AugPOD: Augmentation-oriented Probabilistic Object Detection." CVPR workshop on the Robotic Vision Probabilistic Object Detection Challenge. 2019.
> >
> > **Authors' Final Comment** We hope that the discussion we provided alleviates the reviewer's concerns. We are happy to answer any additional questions the reviewer has.

---

> > > ### Comment · AnonReviewer3 · 2020-11-24
> > > **Good rebuttal**
> > >
> > > Thanks for the rebuttal.  It addresses many of my concerns.  I still think that the paper could have a more focused message, but I agree that the complexity of issues tackled is high.  Thus, writing a simple paper with a focused message may be challenging.  I'm raising my score to 6, and hope to see the paper accepted.

---

### Official Review · AnonReviewer2 · 2020-10-28
**This paper explores a new perspective and has comprehensive experiments but some claims are not convincing enough.**

**Rating:** 6
**Confidence:** 3

**Review:**

Summary:

This paper gives an empirical analysis of the predictive uncertainty for object detection and proposes a tool to establish and evaluate the uncertainty. The main claim of this paper is that, for the bounding box regression, non-local based algorithms have superiority in the consistency with the predictive distributions than local-based methods. The experiments are conducted on some state-of-the-art detectors and fairly demonstrate them.

Strength:

The ablation studies are sufficient and insightful, especially three kinds of state-of-the-art detectors and two important datasets are explored.
The analysis is technical sound and considerably well written.
The performance gain on the transformer based detector (DETR) looks satisfactory.

Weakness:
Section3.3 presents a toy example to show the properties of NLL and the authors claim the weakness of local-based algorithms comes from penalizing lower entropy distributions more severely than higher entropy distributions. But I think this claim is not convincing enough.
The effectiveness of conventional detectors (Faster RCNN and RetinanNet) is relatively limited. Specifically ES has only 0.16 mAP absolute gains over NLL.

---

> ### Author Response · Authors · 2020-11-15
> **Authors' Reply To Reviewer 2**
>
> We thank the reviewer for the insightful comments and constructive criticism.
>
> **Section 3.3 presents a toy example to show the properties of NLL and the authors claim the weakness of local-based algorithms comes from penalizing lower entropy distributions more severely than higher entropy distributions. But I think this claim is not convincing enough. The effectiveness of conventional detectors (Faster RCNN and RetinanNet) is relatively limited. Specifically ES has only 0.16 mAP absolute gains over NLL.**
>
> We agree with the reviewer that on FasterRCNN and RetinanNet, ES does not outperform NLL by a large margin on mAP. We would like to highlight that the weakness in NLL manifests when deterministic backends choose high error regression targets during training, which happens in DETR and to a lesser extent in RetinaNet, but not in FasterRCNN. Since FasterRCNN restricts its regression targets to low error ones during training, there is very little gap in performance of mean estimation shown by mAP between NLL and ES. We modified section 3.3 to convey this point more clearly.
>
> We would also like to highlight that mAP is not a proper scoring rule and does not quantify the quality of predictive distributions estimated by probabilistic object detectors. Figure 2 shows that even though the gap in mAP between NLL and ES is less than 1% on RetinaNet, predictive distributions generated using ES outperform those generated with NLL by a visible margin, a point we discuss in the second paragraph of section 4.2.
>
> **Authors' Final Comments** We hope that the discussion we provided alleviates the reviewer's concerns. We are happy to answer any additional questions the reviewer has.

---

### Official Review · AnonReviewer4 · 2020-10-28
**strong paper with a very good evaluation and takeaways for future research**

**Rating:** 9
**Confidence:** 5

**Review:**

#### Summary
This paper examines the question of how to best estimate the bounding box distribution in probabilistic object detectors, specially regarding the use of non proper scoring rules and issues with the standard negative log-likelihood (NLL, where there is no ground truth for predictive variance).

I believe this paper has many interesting findings that will be useful for the design and training of future probabilistic object detectors. Two main takeaways are that the NLL is not appropriate for training bounding box regressors since it produces high entropy distributions, penalizing overconfident predictions too much. The second message is that the way object detectors select bounding box regression targets with an IoU > 0.5 threshold affects uncertainty in the bounding boxes, since this prevents variations across all IoU ranges to be presented and learned by the detector.

#### Reasons for Score
This paper provides a much needed evaluation of probabilistic object detection from a statistical point of view. Many Computer Vision researchers focus into chasing state of the art metrics (mAP, PDQ, etc) and do not put enough attention into statistical issues such as using non-proper scoring rules, which can hide details and hinder progress.
I believe that this paper will bring a breath of fresh air to the probabilistic object detection field, from its takeaways there are clear directions to improve this kind of detectors and advance the state of the art into other than just chasing metrics. The quality of predictive uncertainty is very important for safe applications of object detection, for example in autonomous driving or robot perception.

#### Pros
- Very good evaluation, with multiple datasets (COCO, OpenImages, and corrupted variations of COCO), producing a robust results and conclusions.
- Good selection of object detectors (DETR, Faster R-CNN, RetinaNet), including one stage and two stage detectors and the recent DETR that proposes a set-based approach to object detection.
- Clear recommendations for future research: use the energy score for training, evaluate with multiple scoring rules, and put attention in the way how bounding box regression targets are selected, with a varied selection of targets being the best, instead of being limited to higher IoU values. We should also rethink the way probabilistic object detectors are evaluated, using proper scoring rules and multiple metrics, since the standard mAP and PDQ can hide details and differences between detectors.
- Using the energy score even for training has a ~2% improvement in mAP over NLL which should also be noted.
- This paper provides a more statistical view on object detection, focused on the bounding box regression problem of a probabilistic object detector, which is not that common in the literature, specially in Computer Vision.
- The paper is well written and a pleasure to read.

#### Cons
- I believe that the energy score might be a bit problematic for object detection, since it requires sampling from a Gaussian, the loss then becomes stochastic (adding noise to the process) and sampling bounding boxes might become a bottleneck during training. This is a minor issue.
- The paper focuses into predictive uncertainty, instead of separating epistemic and aleatoric uncertainty. This is explicitly mentioned in the paper and only a minor issue, since separating both kinds of uncertainty is difficult.

#### Questions for Rebuttal Period
Can you motivate the selection of object detectors? There are many probabilistic object detectors, including Gaussian YOLO for example, so a motivation for these specific detectors would be an improvement.
Can you also motivate the selection of the scoring rules? Why is were the energy score and DMM selected, were other alternatives considered?

#### Minor Issues
- In Section 3.3 and Figure 1, please specify the values of p for DMM and M for the energy score (I assume it is 1000 as mentioned in the Appendix) that are used for evaluation.
- In Figure 3, please add that these results are produced using DETR to the caption, it will be easier to read and interpret.
- This paper has a very statistical view on the object detection problem, maybe it is worth to also take a look at per-class metrics and visualize the produced bounding boxes, some of this analysis could be added to the appendix. Aggregated metrics could also hide details in some classes.
- I think the title could be more informative, since this work focuses into bounding regression uncertainty/variance, this could be part of the title.

---

> ### Author Response · Authors · 2020-11-15
> **Authors' Reply To Reviewer 4**
>
> We would like to thank the reviewer for taking the time to produce such a detailed and thoughtful review.
>
> **Can you motivate the selection of object detectors? There are many probabilistic object detectors, including Gaussian YOLO for example, so a motivation for these specific detectors would be an improvement.**
>
> We thank the reviewer for this feedback, and agree that the paper can be improved by motivating the choice of deterministic detection backends. To summarize, the three deterministic backends are chosen because: 1) They represent one-stage (RetinaNet), two-stage (FasterRCNN), and the recently proposed set-based (DETR) object detectors, providing a good sweep of detection architectures. 2) Open source implementation of DETR, RetinaNet, and FasterRCNN models is publicly available under the Detectron2 object detection framework. Detectron2 is a very reliable, well maintained, and up to date code base which we found to be a key component that enabled our work, and will greatly facilitate the use of our extensions by other researchers. 3) More importantly, Detectron2 has configurations for DETR, RetinaNet, and FasterRCNN with hyperparameters optimized to produce the best detection results for the COCO dataset. The results are also clearly reported on the COCO dataset on Detectron's Github page, which enabled us to know when our deterministic models reached their optimal performance before we added probabilistic capabilities. On the contrary, Gaussian YOLO does not provide results or optimal hyperparameters for the COCO dataset which would render the comparison unfair if it was to be used for our experiments. We have added a summary of this discussion in the third paragraph of Section 4.
>
> **Can you also motivate the selection of the scoring rules? Why were the energy score and DMM selected, were other alternatives considered?**
>
> We thank the reviewer for this feedback. We chose the energy score as a viable proper scoring rule for our experiments due to two reasons: 1) It has theoretical foundation in energy statistics, and can be related to the maximum mean discrepancy, meaning it has the capacity to match all moments of the predicted model distribution to the empirical data distribution. We explain these foundations in Appendix E. 2) It has tractable approximations that are fully differentiable and fast to compute on GPUs, making it a strong candidate target to be used to train neural networks.
>
> We would like to highlight that there are not many proper scoring rules for evaluating continuous valued multivariate predictions in literature. We are currently unaware of any other theoretically founded alternatives to the energy score that can be used to train and evaluate multivariate predictive distributions in a tractable and fast manner. We rewrote the first paragraphs of Section 3.4 to reflect the above two points.
>
> On the other hand, DMM was chosen as it was previously proposed in literature to be used to learn the covariance matrix of Gaussian distributions in object detectors. There are no additional reasons that led to its consideration. We added this point to Section 3.5.
>
> **In Section 3.3 and Figure 1, please specify the values of $p$ for DMM and $M$ for the energy score (I assume it is 1000 as mentioned in the Appendix) that are used for evaluation.**
>
> We thank the reviewer for pointing out this issue. We added the values of $p$ in Section 4 as it varies based on what the deterministic backend uses to compute the bounding box norm loss. The reviewer is correct, we use $M=1000$, which we add to Section 3.4.
>
> **In Figure 3, please add that these results are produced using DETR to the caption, it will be easier to read and interpret.**
>
> We thank the reviewer for pointing out this issue. We modified the caption of Figure 3 to explicitly mention that results are produced using DETR.

---

> > ### Author Response · Authors · 2020-11-15
> > **Cont'd**
> >
> > **This paper has a very statistical view on the object detection problem, maybe it is worth to also take a look at per-class metrics and visualize the produced bounding boxes, some of this analysis could be added to the appendix. Aggregated metrics could also hide details in some classes.**
> >
> > We agree with the reviewer that qualitative images with visualizations improves the paper, and have added qualitative bounding box visualization from DETR and FasterRCNN that shows the classification entropy, the mean bounding box, and the 95% confidence ellipse of bounding box corner predictive distributions for output from models trained with NLL, ES, and DMM losses in Appendix G. We also agree with the reviewer that a detailed exposure of per-category metrics would improve the paper, especially given that some categories such as "toaster" and "hair drier" (Fig C.1) have very few training instances in the COCO dataset. We would like to mention that we have computed the 95% confidence interval around the mean of proper scoring rules over $80$ categories in COCO and showed them as error bars in Figures 2 and 3. We defer a more detailed discussion on per-category performance for future work.
> >
> > **I think the title could be more informative, since this work focuses into bounding regression uncertainty/variance, this could be part of the title.**
> >
> > We thank the reviewer for clearing up a debate that has been ongoing since we began writing this paper. We agree with the reviewer, and modified the title to convey that the paper focuses on regression predictive uncertainty in probabilistic object detectors.
> >
> > **Authors' Final Comments**
> > We hope that the discussion we provided alleviates the reviewer's concerns. We are happy to answer any additional questions the reviewer has.

---

> > > ### Comment · AnonReviewer4 · 2020-11-25
> > > **Good motivation and additional information**
> > >
> > > Thank you for your rebuttal, it addresses most of my questions, I will keep my score and I hope that the paper is accepted. Good work! I believe that this paper will definitely influence the design and training of future probabilistic object detectors and help improve the uncertainty quantification of bounding box regressors, which I think they have ignored by researchers out for some time.

---

### Official Review · AnonReviewer1 · 2020-10-29
**reviews**

**Rating:** 6
**Confidence:** 2

**Review:**

This paper explores the predictive uncertainty estimation problem in object detection. They observe that the commonly used NLL loss leads to high entropy predictive distributions but regardless of the correctness of the output mean. Instead, they use energy score as a non-local proper scoring rule. They also propose an alternative evaluation method.

Strengthens:
1. The paper is well formulated.
2. The authors give a thorough analysis of the experimental results.

Weaknesses and Questions:
I am confused by the main contribution of this paper. Although the authors summary their observations at the end of this paper. I didn't clearly get the differences in the analysis part.
1. No single proper scoring rule can capture all the desirable properties of object detection seems common sense for me. As object detection is a complicated problem that includes both region classification and localization.
2. In Table-1, what does it mean the results of DMM and ES for DETR are much better than NLL , but incremental for RetinaNet and Faster R-CNN?
3. I cannot figure out enough differences between NLL, DMM, and ES from the analysis figures in the experimental part.

I hope the authors can highlight their technical contributions during the rebuttal. I am glad to hear more high-level and straightforward statements for their contributions.

---

> ### Author Response · Authors · 2020-11-15
> **Authors' Reply To Reviewer 1**
>
> We thank the reviewer for the insightful comments and constructive criticism.
>
> **I am confused by the main contribution of this paper. Although the authors summary their observations at the end of this paper. I didn't clearly get the differences in the analysis part. I hope the authors can highlight their technical contributions during the rebuttal. I am glad to hear more high-level and straightforward statements for their contributions.**
>
> We appreciate this feedback, and have made a concerted effort to make the paper more accessible to a wider audience.  We summarize the main points here, and highlight additions to the introduction and discussion in the paper that seek to achieve this end.
>
> To highlight our contributions, we first provide a high level motivation to our paper. Recently, there has been an explosion in papers proposing novel probabilistic object detectors. The majority of these recent approaches for probabilistic object detection conform to a trend that introduces three problems. 1) They approach the probabilistic object detection problem with an end goal of improving state-of-the-art performance on metrics such as mAP and PDQ that are not proper, meaning they do not assess uncertainty predictions correctly. 2) They use negative log likelihood (NLL) as a training loss for learning to output bounding box covariance matrices. 3) They use well-known deterministic detectors such as FasterRCNN, RetinaNet, YOLO, SSD, or others as backends for their probabilistic object detectors without considering the effect of design decisions in these deterministic backends on the probabilistic object detection task.
>
> Our main technical contribution is exposing the pitfalls of this trend, and providing solutions to problems 1 and 2. Specifically:
> 1. We highlight that state of the art metrics such as mAP and PDQ are non-proper scoring rules and do not guarantee a clear ranking of probabilistic object detectors based on the correctness of their estimated predictive distributions. Instead, we provide an evaluation that relies on proper scores such as the NLL, Brier Score, and the Energy score as alternatives to mAP and PDQ for future researchers to use for comparing probabilistic object detectors more reliably.
> 2. We show that using NLL as a training loss can result in bounding box predictive distributions that are of high entropy, meaning they tend to overestimate prediction uncertainty. We relate this phenomenon to the way bounding box regression targets are assigned during training. Unlike FasterRCNN, DETR and RetinaNet choose low IOU (high error) regression targets during training. We find out that NLL biases towards high entropy distributions when used in conjunction with deterministic backends that allow low IOU regression targets such as DETR and RetinaNet.  We propose the Energy Score as an alternative that does not suffer from this problem, and show that it has more accurate predictive distributions when evaluated using proper scoring rules (Figure 2).
> 3. One might suggest that a solution to problem 2 is to restrict regression targets to high IOU ones. We show that this solution is problematic as it restricts the data support used for training variance networks, causing them to fail to learn distributions that have a magnitude of entropy proportional to the magnitude of errors. This behavior is exhibited as the "S" shaped curve of entropy for RetinaNet and FasterRCNN in figure 4, where entropy is seen to not be correctly correlated to the correctness of output detection instances. Ideally, the relation between entropy and detection accuracy would be monotonically decreasing, but this is clearly not the case.
> 4. Finally, we provide clear recommendations for future researchers to avoid the discussed pitfalls in our takeaways section. We added some modifications to the introduction section in hope to clear up our contributions.
>
> **No single proper scoring rule can capture all the desirable properties of object detection seems common sense for me. As object detection is a complicated problem that includes both region classification and localization.**
>
> We agree with the reviewer that it should be common sense that no single proper scoring capture all the desirable properties of complicated problems such as object detection. However, as explained above, the trend in literature seems to be the usage of a single *non-proper* scoring metric such as mAP or PDQ to jointly evaluate both regression and classification tasks in probabilistic object detectors. Our conclusion also implies that it is essential to *use both local and non-local proper scoring* rules to cover each other's weaknesses in both classification and detection tasks. As such, our conclusion highlights a more reliable and precise way of assessing probabilistic object detectors for future researchers.

---

> > ### Author Response · Authors · 2020-11-15
> > **Cont'd**
> >
> > **In Table-1, what does it mean the results of DMM and ES for DETR are much better than NLL, but incremental for RetinaNet and Faster R-CNN?**
> >
> > In the first paragraph of the subsection titled "Pitfalls of Training and Evaluation Using NLL" we discuss how NLL performs poorly when regression targets with low IOU with ground truth are selected during training. Figure 4 shows that DETR continuously selects low IOU targets during training and as such DETR with NLL performs worse than DETR with ES or DETR with DMM. On the other hand, FasterRCNN restricts regression targets to have a high IOU with ground truth during training. As such, we observe NLL to produce lower mAP values when used to train DETR, but maintains the same mAP as ES when used to train FasterRCNN. We added this explanation to the "Pitfalls of Training and Evaluation Using NLL" section(page 8) in the paper.
> >
> > **I cannot figure out enough differences between NLL, DMM, and ES from the analysis figures in the experimental part.**
> >
> > We would like to highlight that not all figures are meant to measure the performance gap between NLL, DMM, and ES. Figure 3 is to show that it is important to use the Brier score as well as NLL for evaluating the quality of predictive distributions of the classification subtask of probabilistic object detectors. Figure 5 shows that variance networks can differentiate between in-distribution and out-of-distribution instances in probabilistic object detectors. The first column of figure 2 shows the MSE to demonstrate that the difference in regression performance with proper scoring rules is due to differences in bounding box covariance matrix estimates and not mean estimates. We reduced the clutter by removing redundant legend boxes and improved the y-axis scale to better highlight the differences between NLL, ES, and DMM.
> >
> > **Authors' Final Summary**
> > In summary, we discovered three problems with the trend followed by the state of the art probabilistic object detectors. We provided solutions for two of the three problems, and left one as future work. Our work aims to induce a shift in the highlighted problematic trend of the research community, a shift that will not happen on its own as we see that probabilistic object detectors that optimize for mAP [1] or PDQ [2], use NLL as a training loss [1] and use regression target selection restricted to high IOU targets [1] continue to appear in literature after the submission of our paper in October. We hope that our discussion highlights the contributions of our paper within the domain of probabilistic object detection. We are happy to answer any additional questions the reviewer has.
> >
> > [1] Zhong, Yuanxin, Minghan Zhu, and Huei Peng. "Uncertainty-Aware Voxel based 3D Object Detection and Tracking with von-Mises Loss." arXiv preprint arXiv:2011.02553 (2020).
> >
> > [2] Azevedo, Tiago, René de Jong, and Partha Maji. "Stochastic-YOLO: Efficient Probabilistic Object Detection under Dataset Shifts." arXiv preprint arXiv:2009.02967 (2020).

---

### Decision · Program_Chairs · 2021-01-07
**Final Decision**

**Decision:**

Accept (Poster)

**Comment:**

The initial reviews were mixed (2 positive, 2 negative).  The main concerns were about presentation issues: unclear contribution or main point; unclear analysis of figures; missing some motivation of selecting object detectors; etc.).  On the other hand, reviewers appreciated the well-formulated paper, analysis and recommendations from the experiments;

The author response addressed the presentation issues and added additional motivations and clarifications. All reviewers in the end recommended accept.